# Response of downstream lakes to Aru glacier collapses on the western Tibetan Plateau

Yanbin Lei[1,2*], Tandong Yao[1,2], Lide Tian[2,3], Yongwei Sheng[4], Lazhu[5], Jingjuan Liao[6], Huabiao Zhao[1, 2,10], Wei Yang[1, 2], Kun Yang[2, 7], Etienne Berthier[8], Fanny Brun[9], Yang Gao[1,2], Meilin Zhu[1], Guangjian Wu[1, 2]

[1]Key Laboratory of Tibetan Environment Changes and Land Surface Processes, Institute of Tibetan Plateau Research, Chinese Academy of Sciences, Beijing 100101, China

[2]CAS Center for Excellence in Tibetan Plateau Earth System, Beijing, 100101, China

[3]Institute of International Rivers and Eco-security, Yunnan University, Kunming, China

[4]Department of Geography, University of California, Los Angeles (UCLA), CA 90095–1524, USA

[5]National Tibetan Plateau Data Center, Institute of Tibetan Plateau Research, Chinese Academy of Sciences, Beijing 100101, China

[6]Key Laboratory of Digital Earth Science, Institute of Remote Sensing & Digital Earth, Chinese Academy of Sciences, Beijing 100094, China

[7]Department of Earth System Science, Tsinghua University, Beijing 10084, China

[8]LEGOS, CNES, CNRS, IRD, UPS, Université de Toulouse, Toulouse, France

[9]Université Grenoble Alpes, CNRS, IRD, Grenoble INP, IGE, 38000 Grenoble, France

[10]Ngari Station for Desert Environment Observation and Research, Institute of Tibetan Plateau Research, Chinese Academy of Sciences, 100101, China

*Correspondence to*: Yanbin Lei (leiyb@itpcas.ac.cn)

**Abstract** The lower parts of two glaciers in the Aru range on the western Tibetan Plateau (TP) collapsed on 17 July and 21 September 2016, respectively, causing fatal damage to local people and their livestock. The giant ice avalanches, with a total volume of $150 \times 10^6$ m$^3$, had almost melted by September 2019 (about 30% of the second ice avalanche remained). The impact of these extreme disasters on downstream lakes has not been investigated yet. Based on in-situ observation, bathymetry survey and satellite data, we explore the impact of the ice avalanches on the two downstream lakes (i.e. Aru Co and Memar Co) in terms of lake morphology, water level and water temperature in the subsequent four years (2016-2019). After the first glacier collapse, the ice avalanche slid into Aru Co along with a large amount of debris, which generated great impact waves in Aru Co and significantly modified the lake's shoreline and underwater topography. An ice volume of at least $7.1 \times 10^6$ m$^3$ was discharged into Aru Co, spread over the lake surface and considerably lowered its surface temperature by 2-4 $^{\circ}$C in the first two weeks after the first glacier collapse. Due to the large amount of meltwater input, Memar Co exhibited more rapid expansion after the glacier collapses (2016-2019) than before (2003-2014), in particular during the warm season. The melting of ice avalanches was found to contribute to about 23% of the increase in lake storage between 2016 and 2019. Our results indicate that the Aru glacier collapses had both short-term and long-term impacts on the downstream lakes and provide a baseline in understanding the future lake response to glacier melting on the TP under a warming climate.

## 1 Introduction

Potential risk of natural hazards in the Third Pole region has increased in the last decades (Cui et al., 2015; Wang et al., 2018; Cook et al., 2018; Liu et al., 2019). Glaciers in this region have changed heterogeneously due to rapid climate warming and different patterns of precipitation changes (Yao et al., 2012a). Most glaciers have experienced significant negative mass balance, except for the slight mass gain in the Karakoram and western Kunlun Mountains (Kääb et al., 2015; Brun et al., 2017; Shean et al., 2020). Due to the rapid glacier retreat, most glacial lakes expanded rapidly and many new glacial lakes appeared (Li et al., 2012; Nie et al., 2017; Shugar et al., 2020), which increased the risk of glacial lake outburst floods (Cook et al., 2018; Wang et al., 2018). Meanwhile, as a new form of glacier instability, ice avalanche appeared on the western Tibetan Plateau. The lower parts of two glaciers in the Aru range on the western TP collapsed on 17 July and 21 September 2016 (i.e. Aru-1 and Aru-2 glacier collapses), leading to fatal damage to local people and livestock (Tian et al., 2017). Together with soft-bed properties of the glaciers, unusually high water input from melting and precipitation was identified as one of the main causes of the two glacier collapses (Kääb et al., 2018; Gilbert et al., 2018).

Although the two giant ice avalanches have caused serious ecological and environmental problems, their impact on the downstream lakes (i.e. the outflow lake Aru Co and the terminal lake Memar Co) in the subsequent years has not been investigated yet. The two ice avalanches may influence the downstream lakes in several ways. First of all, a large amount of ice slid into Aru Co at high speed and generated huge impact waves (Kääb et al., 2018), which could affect shoreline and underwater topography. Secondly, the melting of the fragmented ice mass, with a total volume of ~$150 \times 10^6$ m$^3$, could supply additional water to the downstream lakes and affect lake level changes of Memar Co in the subsequent years. Thirdly, the ice avalanches could affect lake surface temperature through the spread of ice floes and cold water input. In fact, there are many studies about the impact of glacier melting on rapid lake growth on the interior TP (e.g. Lei et al., 2012; Zhou et al., 2019; Treichler et al., 2019). However, the process of glacier meltwater regulating lake water balance is largely unknown due to a lack of in-situ observation. Therefore, the observation of lake level changes in the downstream lakes of the Aru glacier collapses not only provides us unique evidence of the impact of a large amount of glacier melting on the downstream lakes, but also is essential to understand the relationship between glacier mass loss and lake behaviour on the TP under a warming climate.

In September 2016, two months after Aru-1 glacier collapse and one week after Aru-2 glacier collapse, we conducted a field campaign and installed instruments to monitor lake level changes at the two downstream lakes, Aru Co and Memar Co (Fig. 1). In July 2017 and October 2018, we further conducted bathymetry survey at both lakes. A comprehensive hydro-meteorology monitoring was established in the two downstream lakes in October 2018. This study explores the impact of the two glacier collapses on the downstream lakes in terms of lake morphology, lake level changes and lake surface temperature. We first investigate the instantaneous impact of the Aru-1 ice avalanche on the morphology of Aru Co, then evaluate the impact of the meltwater on lake level changes at Memar Co on seasonal to inter-annual scales, and finally analyse the impact of the meltwater on lake surface temperature (LST) at both lakes.

## 2 Study area

Aru Co and Memar Co, located in an endorheic basin on the western TP (Fig. 1), are the two lakes in the downstream of the glacier collapses. According to the second Chinese glacier inventory (Guo et al., 2015), 105 pieces of glaciers are located in the basin covering a total area of ~184 km$^2$. Glaciers in this region have been rather stable in the past four decades before the two ice avalanches (Tian et al., 2017; Zhang et al., 2018). Two adjacent glaciers (Aru-1 and Aru-2) to the west of Aru Co collapsed suddenly on 19 July and 21 September, 2016, respectively, killing nine people and hundreds of livestock. The Aru-
1 ice avalanche reached Aru Co at high speed after running out 6-7 km beyond the glacier terminus, generating huge impact waves at Aru Co (Kääb et al., 2018). On the Aru-1 glacier collapse fan, the thickness of the fragmented ice mass varied from 3 m at the glacier snout to 13 m at the far end of the deposit (Tian et al., 2017). The two ice avalanches covered an area of 9.4 and 6.7 km$^2$, and the detached volumes were estimated to be 68 and 83×10$^6$ m$^3$, respectively (Tian et al., 2017; Kääb et al., 2018).

Aru Co and Memar Co are lagoons and share a catchment area of 2310 km$^2$. Aru Co is an outflow lake with salinity of 0.56 g/L, and Memar Co is the terminal lake of Aru Co with salinity of 6.22 g/L. The surface elevation of Aru Co (4937 a.s.l.) was about 14 m higher than Memar Co (4923 a.s.l.) in 2003, according to ICESat satellite altimetry data (Li et al., 2014). There are dozens of visible paleo-shorelines around Memar Co. The highest shoreline around Memar Co is ~40 m above the modern lake level, indicating Aru Co and Memar Co used to be one large lake on a geological time scale.

The climate in this area is cold and dry most of the year. Automatic weather station (AWS) data collected between October 2016 and September 2019 near the glacier collapse (~5000 a.s.l.) show that mean annual air temperature is -3.6 $^o$C, with the lowest value in January (-14.0 $^o$C) and the highest value in August (7.2 $^o$C). A T200B rain gauge data indicates that mean annual precipitation near the collapse fan is 333 mm between October 2016 and September 2019, which is much higher than that at Nagri meteorological station (Tian et al., 2017). More than 80% of annual precipitation in this region is concentrated
in the warm season from June to September. Snowfall in the cold season between October and May only accounts for 10-15% of the annual precipitation.

>>Fig. 1<<

## 3 Methods

### 3.1 Lake bathymetry

Bathymetric surveys at Aru Co and Memar Co were conducted in July 2017 and October 2018, respectively. Water depth was determined using a 500 Watt duel frequency depth sounder interfaced with a Garmin GPSMAP 421S chart plotter. Latitude, longitude, and water depth were recorded at 3-second interval during each bathymetric survey. At Aru Co, a total of 16,100 water depth points were acquired. A detailed bathymetry survey near the first glacier collapse fan was conducted at an interval of 100-200 m at Aru Co. At Memar Co, a total of 18,000 water depth points were acquired. The horizontal

position of each point was recorded with an accuracy of 3 m or better. The lake boundary in July 2017 and October 2018 was used to calculate lake water storage at Aru Co and Memar Co, respectively. The water depth was interpolated to the whole lake to acquire the lake isobaths and then lake volume was calculated in ArcGIS 10.1. Lake water depth of the 1994 shoreline was reconstructed according to bathymetry survey and used to calculate the lake isobaths because the lake level in middle to late 1990s was much lower than today (Fig. S1) and Memar Co was composed of two lakes at that time.

Uncertainty of the lake volume is estimated to be ±6% by comparing the reconstructed lake level and satellite altimetry between 2004 and 2018.

### 3.2 Lake water level monitoring

Lake level at Aru Co and Memar Co was monitored since September 2016 using HOBO (U20-001-01) or Solinst water level loggers, which were installed in the littoral zone of the lake. Because water levels were recorded as changes in pressure (less

than 0.5 cm water level equivalent), air pressure data was subtracted from the level loggers to get pressure changes related to water column variations. Daily lake level changes between October 2016 and September 2019 were used in this study at Aru Co. At Memar Co, lake level is only available from October 2017 to September 2019 due to the loss of the logger in the first year. Water depth of the loggers was measured during fieldwork to calibrate the logger data.

### 3.3 Satellite observation

Multi-sources of satellite data, including Landsat images, ICESat and CryoSat-2 satellite altimetry, were explored to detect long-term changes in lake extent and water level. Landsat images downloaded from the USGS website (http://glovis.usgs.gov) were used to investigate changes in lake area since the 1970s. A total of 30 satellite images between September and November, 1972 to 2018, were selected. Before 1990, only two images (1972 and 1976) were available. After 1990, almost annual changes in lake area (no data in 1991, 1995 and 1998) were extracted. Lake boundaries were

extracted in false colour image by manual delineation using ArcGIS 10.1 software. The accuracy of manual digitization is controlled within an error of one pixel (30 m).

ICESat and CryoSat-2 satellite altimetry data was used to detect lake level changes between 2003 and 2017. ICESat altimetry data was processed after Li et al (2014) and was used to examine water level variations between 2003 and 2009. CryoSat-2 data was processed after Xue et al (2018) and was used to investigate water level variations between 2010 and

2018. Both lakes were observed by ICESat satellite twice or three times a year (Phan et al., 2012), and by CryoSat-2 satellite every two or three months (Kleinherenbrink et al., 2015; Jiang et al., 2017). Notably, the two datasets are referenced to different ellipsoids and geoid height. The ICESat data contains corrected surface ellipsoidal heights referenced to TOPEX/Poseidon ellipsoid and geoid height referenced to Earth Gravity Model (EGM) 2008; while the CryoSat-2 data are referenced to WGS84 and EGM96 (Song et al., 2015). At Aru Co, the lowest lake level in May is very stable from year to

year as it is controlled by the elevation of the outlet. The ICESat and CryoSat-2 derived lake surface elevations of Aru Co were averaged to be 4936.67 m a.s.l. in April (n=2) during the period 2003-2009 and 4937.04 m a.s.l. in May (n=5) during

the period 2011-2016, respectively. The elevation difference of 0.37 m was determined as the bias of the two datasets and used to correct satellite altimetry data.

Dynamics of the two ice avalanche deposits were investigated through different kinds of satellite images (Sentinel-2,
Gaofen-2, Landsat-8 OLI, SPOT7, Pléiades and ASTER DEMs). A Sentinel-2 satellite image on 21 July 2016 (4 days after the Aru-1 glacier collapse) was acquired to detect the extent of the intruding ice into Aru Co. A high resolution (~1 m) Gaofen-2 satellite image on 25 July 2016 was acquired to detect the ice floes over the surface of Aru Co. The extent of the two ice avalanches was extracted based on Landsat images between 2016 and 2019 and used to calculate meltwater every year. Meltwater originating from the avalanche deposits is constrained using examination of satellite images and differencing
of digital elevation models (DEMs). SPOT7, Pléiades and ASTER DEMs are calculated using the Ames Stereo Pipeline (Shean et al., 2016) using processing parameters from recent studies (Berthier and Brun, 2019; Dussaillant et al., 2019; Miles et al., 2018). DEMs are coregistered using the method described in Berthier et al. (2007).

### 3.4 Long-term lake level reconstruction

The outflow lake Aru Co has been very stable since the 1970s, while the terminal lake Memar Co has expanded significantly
since the 2000s. Before 2003, lake level variations at Memar Co were determined based on the current lake bathymetry and the position of past shorelines, which were derived from Landsat satellite images. Bathymetric survey was used to determine the current water depth over shorelines that were previously exposed (Lei et al., 2012). To minimize errors, more than 10 bathymetry transects across Memar Co were acquired and used to reconstruct past lake level changes. Lake level changes in 1972, 1994, 1999, 2004 and 2014 relative to October 2018 were reconstructed. Here we assume that lake level changes were
constant across the lake and that the uncertainty of lake level changes was determined as the standard deviation of all the reconstructed lake levels. In this study, uncertainty of changes in lake level was estimated to be ±0.6 m in the 1970s and ±0.3 m since the 1990s. The relationship between lake area and the reconstructed lake levels was developed using a second order polynomial regression model. Continual lake level changes at Memar Co since 1972 were reconstructed using this relationship and the corresponding lake area.

### 3.5 Lake surface temperature derived from MODIS satellite data

MODIS 8-day land surface temperature products (i.e. Terra-MOD11A2 and Aqua-MYD11A2) were used to investigate changes in lake surface temperature at Aru Co and Memar Co. In both platforms (Terra and Aqua), two instantaneous observations were collected every day (Terra: approximately 10:30 and 22:30 local time, Aqua: approximately 13:30 and 01:30 local time). The MODIS 8-day data is the averaged lake surface temperature of daily MODIS product over eight days.
We used the latest version of MODIS products (V006) in this study. Only nighttime data were used because there was less cloud cover at night (Zhang et al., 2014; Wan et al., 2018). MOD11A2 and MYD11A2 products are available at a spatial resolution of about 1 km with an accuracy of 1 K under clear sky conditions (Wan Z., 2013). MODIS lake surface temperature data is pre-processed to account for atmospheric and surface emissivity effects. The cloud mask (MOD35) used

for inland water provides a surface temperature measurement when there is a 66 % or greater confidence of clear-sky conditions (Wan Z., 2013), otherwise no temperature is produced. To reduce the contamination from land pixels, only lake pixels beyond 1 km from shoreline were extracted (Ke et al., 2014). At Aru Co, lake surface temperature at the southern and northern (closer to the ice avalanches) halves of the lake was extracted to investigate its spatial difference (Fig. 2). At Memar Co, lake surface temperature at the northern half of the lake was extracted. Anomalous lake surface temperature was examined and removed if there was big difference between MOD11A2 and MYD11A2 datasets. To confirm the reliability of MODIS products, nighttime lake surface temperature was compared with in-situ observation at the shoreline.

## 4 Results

### 4.1 Bathymetry survey at Aru Co and Memar Co

Aru Co has a surface area of ~105 km$^2$ with a length of 27 km and a width of 1.4 to 9 km. Observation from satellite images shows that Aru Co is composed of two sub-basins. The northern basin accounts for less than 30% of the total lake area with a maximum water depth of 20 m. The southern basin is the main body of Aru Co, with a maximum water depth of 35 m (Fig. 2). The central part of Aru Co is narrow and shallow, with a width of ~1.5 km and a maximum water depth of ~11 m. The entire Aru Co has an average water depth of 17.6±1.0 m and total water storage of 1.8±0.1 Gt.

Memar Co has a surface area of 177 km$^2$ in 2018 with a length of 36 km and a width of 2 to 7 km. Similar to Aru Co, Memar Co is also composed of two sub-basins. The northern part is the main water body of the lake with a maximum depth of 42.6 m. The southern part accounts for less than 20% of total lake area, with a maximum water depth of 20.5 m (Fig. 2). The south-central part of Memar Co is narrow and shallow, with a width of 2-3 km and a maximum depth of ~12.5 m. Satellite images show that the southern and northern parts were separated in the 1990s when the lake level was lower than today (Fig. S1). The two parts have been connected again since 2000 due to the rapid lake expansion. According to lake bathymetry in October 2018, Memar Co has an average water depth of 20.0±1.2 m and total water storage of 3.5±0.2 Gt, about twice as large as Aru Co.

>>Fig. 2<<

### 4.2 Instantaneous impact of Aru-1 glacier collapse on the Aru Co morphology

Aru-1 glacier collapse ran into Aru Co at high speed after running out 6-7 km beyond the glacier terminus (Tian et al., 2017; Kääb et al., 2018). A Sentinel-2 satellite image acquired on 21 July 2016 (4 days after the collapse) showed that the ice avalanche ran into Aru Co as far as ~800 m and the intruding ice had an area of ~0.89 km$^2$ with a width of ~2250 m and an average length of 400 m. The intruding ice generated great waves at the northern Aru Co due to its high speed and large volume, which inundated the opposite shore of Aru Co (Kääb et al., 2018). Fieldwork in October 2016 showed that there was clear footprint of wave erosion at the opposite shore of the northern Aru Co, which extended up to 240 m inland and 9 m above the lake level along a 10 km section of the shoreline (Fig. 3a).

The bathymetry survey in July 2017 showed that water depth at the east margin of the intruding ice was about 8 m. Because the intruding ice was obviously higher than the lake surface, 8 m is a lower bound for the ice thickness. Therefore, the volume of ice mass entering Aru Co is estimated to be at least $7.1 \times 10^6$ m$^3$, accounting for ~10% of the total ice volume of Aru-1 glacier collapse. Due to the influence of lake water, the intruding ice melted quickly in less than two months as observed by Landsat satellite image on 20 September 2016 (Fig. 4).

Detailed bathymetry survey at Aru Co showed that the underwater topography near Aru-1 ice avalanche was largely modified due to a large amount of debris input along with the fragmented ice mass. Fig. 3b shows that the bathymetry near the ice avalanche became uneven, in contrast with the adjacent areas. The uneven underwater topography indicated that a large amount of debris was transported into Aru Co or the lake bed was significantly eroded. The extent of the uneven lake bathymetry was slightly larger than that of the intruding ice on 21 July 2016 (Fig. 3b), indicating that part of the intruding ice

had spread over the surface of Aru Co or melted in four days after the glacier collapse. The lake bottom stays unchanged (i.e. smooth) in areas deeper than 15 m or far from the glacier collapse fan.

An investigation of Aru-1 ice avalanche fan in October 2019 gave further evidence of debris input into Aru Co. Clear deposit with a thickness of 0.2-1.0 m  was left after the fragmented ice mass had completely melted. The original road was no longer accessible because of a thick debris cover. Boulders with a diameter of 1-2 m were found even near the shoreline (Fig. 3c).

The uneven land surface may explain how the lake bottom became uneven. Fieldwork also showed that Aru Co's shoreline near the northern and southern sides of the ice avalanche moved offshore by about 100-120 m, which was probably due to the deposit of debris transported by glacier collapse and afterwards by meltwater.

>>Fig. 3<<

>>Fig. 4<<

**4.3 Estimation of meltwater from the two ice avalanches**

According to the areas and volumes reported by Kääb et al (2018), the average thicknesses of Aru-1 and Aru-2 ice avalanche deposits in October 2016 were 7.6 m and 15.2 m, respectively. The contrasting thicknesses of the fragmented ice masses led to different duration of their melting. Aru-1 glacier collapse had almost melted in two summers as indicated by satellite images in October 2017 (only some scattered ice mass remained). The elevation difference derived from a pre-collapse

SPOT7 DEM (25 November 2015) and a Pléiades DEM (28 August 2018) indicates an elevation change of almost zero (-0.10±0.50 m) over the area of Aru-1 avalanche deposit, so the entire ice mass melted away by August 2018 (Fig. S2). The melting of Aru-2 glacier collapse lasted longer due to its larger thickness. In October 2019, the fragmented ice had an area of about 1.9 km$^2$, accounting for about 29% of the initial area. The remaining ice mass mainly occurred in the upper part of Aru-2 ice avalanche, where the fragmented ice was thicker (Kääb et al., 2018). The elevation difference derived from

ASTER DEMs between November 2011-2012 and January 2020 indicates that about $31.8 \pm 3.8 \times 10^6$ m$^3$ of ice and debris remained over the area of the Aru-2 ice avalanche deposit (Fig. S3), corresponding to about 30% of its initial volume.

Here we roughly estimate the annual meltwater of the fragmented ice mass according to the area and in-situ measurements of ice mass balance. We do not consider sublimation/evaporation or other kinds of water loss in this study because they are relatively small and negligible (Li et al., 2019) compared with the rapid melting of the ice avalanches. In-situ measurements at 9 sites showed that Aru-1 ice avalanche thinned about 2.84 m on average between 13 August 2016 and 24 October 2016, which corresponded to an ice mass volume of $25.4 \times 10^6$ m$^3$ (Tab. 1). Considering the intruding ice into Aru Co ($7.1 \times 10^6$ m$^3$), the total meltwater of the first ice avalanche was estimated to be $29.2 \times 10^6$ m$^3$ in 2016 (assuming the ice density of 900 kg/m$^3$). The meltwater of Aru-2 ice avalanche was not considered in 2016 because air temperature was already close to 0 $^o$C in early October. The largest melting of the fragmented ice mass occurred in summer 2017. In-situ measurements show Aru-1 and Aru-2 ice avalanches melted down 6.5 m and 5.5 m on average, respectively, between September 2016 and September 2017. Almost all the Aru-1 ice avalanche had melted by October 2017 and the meltwater was estimated to be $35.5 \times 10^6$ m$^3$ in 2017, which is also the remaining part of Aru-1 ice avalanche. Meltwater of the Aru-2 ice avalanche was estimated to be $31.2 \times 10^6$ m$^3$. Thus, the cumulative meltwater during the period 2016-2017 reached $60.0 \times 10^6$ m$^3$. In October 2018 and 2019, the Aru-2 ice avalanche deposit only had an area of 3.0 and 1.9 km$^2$, respectively. We assumed that the melt rate of the ice deposit in 2018 and 2019 were same as in 2017, and the total volume of meltwater was estimated to be $19.2 \times 10^6$ m$^3$ and $12.0 \times 10^6$ m$^3$ in 2018 and 2019, respectively (Tab. 1). Thus, about $17.2 \times 10^6$ m$^3$ of the ice remained for Aru-2 ice avalanche by October 2019 according to the above calculation, which is generally consistent with the remaining volume estimated by differencing ASTER DEMs.

>>Tab. 1<<

## 4.4 Impact of the meltwater on the seasonal lake level changes of Memar Co

The meltwater had more impact on Memar Co's lake level changes than Aru Co's because Aru Co is an outflow lake. We first investigated the seasonal lake level changes at both lakes and their hydraulic connections based on in-situ observations and satellite altimetry data between 2016 and 2019. Aru Co exhibited distinct seasonal fluctuations with the lowest lake level in late May and the highest in late August (Fig. 5). Its lake level usually increased by 30-50 cm between June and August in response to the relatively high summer rainfall and glacier runoff. After the end of monsoon rainfall, the lake level decreased considerably by 20-30 cm due to river discharge and lake evaporation between September and October. When the lake surface was frozen between November and the following April, the lake level exhibited a slight drop by 10-15 cm. After the lake ice broke up in early May, the lake level of Aru Co continued to decrease slightly due to very limited runoff and low evaporation.

>>Fig. 5<<

Compared to Aru Co, the lake level at Memar Co did not exhibit distinct seasonality during the study period. There was an overall lake level increase throughout the year. Lake level increase not only occurred in the warm season, but also in the cold season (Lei et al., 2017). During the cold season, lake level increased by ~30 cm (i.e. 1.4-2.0 mm/day) between November and May, which was comparable or even larger than that in the warm season between June and August (Fig. 5). The rate of

lake level increase in the cold season was very stable, indicating a steady water supply as well. Lake level increase in the warm season was mainly associated with high summer rainfall and glacier melting, while the lake level increase in the cold season was probably related to groundwater discharge because there is almost no surface runoff during this period. Notably, the lake volume decrease at Aru Co only accounted for 20-30% of the lake volume increase at Memar Co during the ice covered season (November to May), indicating that the significant lake water surplus at Memar Co was not mainly contributed by the discharge from Aru Co. The in-situ observation of seasonal lake level changes at Memar Co confirms the significant lake level increase in cold season on the western Tibetan Plateau (Lei et al., 2017).

The hydraulic connection between Aru Co and Memar Co is investigated by comparing the seasonal lake level changes at the two lakes. Lake level at Aru Co started to increase rapidly in early July, which was about half a month earlier than that at Memar Co. Meanwhile, the end of the rapid lake level increase at Aru Co was also about half a month earlier than that at Memar Co (Fig. 5b, c). The time lag of seasonal lake level changes at the two lakes indicates the buffering effect of Aru Co as an outflow lake. A large amount of water was stored in Aru Co in summer, and released to Memar Co in autumn. In early September, lake level at Aru Co decreased by about 10 cm, accounting for about 90% of the lake volume increase at Memar Co. This indicates that Aru Co, as an outflow lake, plays a significant role in regulating the water balance of Memar Co.

The impact of the two glacier collapses on lake level changes can be found from the seasonal lake level changes derived from CryoSat-2 satellite data and in-situ observations between 2011 and 2019. The lake level increase in cold season (October to May) did not vary much from year to year, with an average value of 0.35 m before (i.e. 2011-2015) and 0.36 m after (i.e. 2016-2019) the glacier collapses (Fig. 6a). However, lake level increase in the warm season (May to September, refer to as 'summer' in the following) increased drastically after the glacier collapses (Fig. 6b). Before the glacier collapses, lake level increase in the warm season varied in a range of -0.2~0.36 m, with an average of 0.12 m. After the glacier collapses, the lake level increase in the warm season varied in a range of 0.24~0.54 m, with an average of 0.39 m. Since the fragmented ice mass mainly melted in summer, the contribution of meltwater to the lake level increase in summer was estimated to be 44.9% on average between 2016 and 2019 by comparing the meltwater and lake storage increase at Memar Co. We can see that the melting of the fragmented ice mass played an important role in the lake level increase in summer at Memar Co.

>>Fig. 6<<

## 4.5 Impact of the meltwater on the inter-annual lake level changes of Memar Co

Lake level changes at Memar Co between 1972 and 2018 were investigated based on satellite altimetry data and bathymetry survey. Lake level changes between 2003 and 2018 were observed by ICESat and CryoSat-2 satellite altimetry. Earlier lake level changes before 2003 were reconstructed according to the bathymetry survey and past shorelines (Lei et al., 2012). Bathymetry survey showed that the lake level of Memar Co was 10.4±0.6 m, 12.3±0.3 m, 12.5±0.3 m, 8.3±0.3 m, 3.1±0.3 m lower in 1972, 1994, 1999, 2004 and 2014 relative to October 2018. According to the lake area and the corresponding water

level in the five years, the relationship between lake area and water level was developed by using a second order polynomial regression ($R^2 = 0.99$):

$$y = -0.1077 \times x^2 + 2.8468 \times x + 176.81$$

Here, y is lake area ($km^2$), and x is lake water level (m). Thereby, we reconstructed lake level changes since 1972 by using the corresponding lake area (Fig. 7). To validate the results, we compare the reconstructed lake level changes with satellite altimetry data during the overlap period of 2003-2018. Fig. 7a shows that there is a good correspondence between the two datasets, indicating our reconstructed lake level changes are reliable.

>>Fig.7<<

Based on lake area and water level changes, lake dynamics of Memar Co between 1972 and 2018 can be divided into two distinct periods (Fig. 7). Between 1972 and 1999, Memar Co exhibited gradual shrinkage with lake level decrease of 2.1±0.3 m. Since 2000, Memar Co experienced rapid expansion with lake level increase of 12.5±0.3 m between 2000 and 2018. The gradual shrinkage before 1999 and rapid expansion since 2000 at Memar Co were similar to most endorheic lakes on the TP (e.g. Lei et al., 2014). Many studies showed that precipitation increased significantly on the interior TP since the late 1990s (Yang et al., 2014; Treichler et al., 2019), which led to the significant lake expansion (Lei et al., 2014). Between 1972 and 2018, lake level and water storage of Memar Co increased by 10.4±0.6 m and 1.62±0.11 Gt (from 1.86 to 3.49 Gt), respectively.

After the Aru glacier collapses in 2016, Memar Co expanded more rapidly than before. Between 2003 and 2014, the lake level of Memar Co increased steadily at a rate of 0.59 m/yr. The lake expansion paused in 2015, which was probably in response to the widespread drought over the TP during the strong 2015/2016 El Niño event (Lei et al., 2019). Between 2016 and 2019, the lake level of Memar Co increased at an average rate of 0.80 m/yr, about 30% higher than that between 2003 and 2014. During this period, the lake level and the water storage of Memar Co increased by 3.0 m and 0.52 Gt, respectively. Assuming all the meltwater was transferred into Memar Co (after transiting through Aru Co), the total melting of ice avalanches contributed to 23.3% of increase in lake storage between 2016 and 2019. Without the additional meltwater of ice avalanches, the rate of lake level increase of Memar Co after the glacier collapses could be similar to that between 2003 and 2014 (Fig. 7a).

The contribution of the melting of ice avalanches on inter-annual lake level changes of Memar Co was also quantitatively evaluated. In 2016, when ice melting mainly occurred in the first glacier collapse, Memar Co expanded slightly with lake level increase of 0.43 m. In 2017, when the ice melting reached the maximum, Memar Co exhibited the most rapid expansion, with lake level increase of 1.07 m. In 2018 and 2019, with the decrease of the meltwater from ice avalanches, the expansion of Memar Co also slowed down, with lake level increase of 0.8 m and 0.69 m, respectively. Assuming all the meltwater was transferred into Memar Co, its contribution to the lake level increase of Memar Co was estimated to be 40.0%, 32.2%, 13.6% and 9.9% in the 4 years of our study period (2016-2019).

## 4.6 Impact of Aru glacier collapses on lake surface temperature

Seasonal variations of lake surface temperature at Aru Co and Memar Co are shown in Fig. 8. Aru Co usually freezes up in early November and breaks up in early May, with lake ice phenology for about six months. After lake ice breaks up in May, the nighttime lake surface temperature increases rapidly from 2 $^{o}$C to 10 $^{o}$C between May and August. Then the lake water cools gradually from September to October. Seasonal lake surface temperature at Memar Co shows similar seasonal cycle with Aru Co, but delayed lake ice phenology (Fig. 8). Memar Co usually freezes up in late November and breaks up in early June. A comparison of MODIS derived lake surface temperature with in-situ observations shows that although there are similar seasonal cycles, in-situ lake surface temperature at the shoreline is considerably higher than MODIS derived lake surface temperature in spring and summer (Fig. S5). This is because MODIS sensors measured the lake skin temperature at the lake centre while HOBO logger measured lake water temperature at the depth of 30-70 cm at the shoreline.

The Aru-1 ice avalanche significantly affected lake surface temperature of Aru Co and Memar Co in summer 2016. Both Terra and Aqua datasets showed that nighttime lake surface temperature at Aru Co decreased abruptly by 2-4 $^{o}$C in the first two weeks after the Aru-1 glacier collapse (Fig. 8), which was quite different from normal years and may considerably affect the lake ecosystem. A similar decrease in lake surface temperature also occurred at Memar Co, but with less magnitude and shorter duration than at Aru Co (Fig. 8). We attribute the significant decrease in lake surface temperature to the ice floes over the surface of Aru Co. As shown in Section 4.2, a large amount of ice mass slid into Aru Co after Aru-1 glacier collapse and generated great waves at Aru Co. High resolution (1 m resolution) Gaofen-2 satellite image on 25 July 2016 showed a large amount of ice floes spread over the surface of Aru Co (Fig. 4). The gradual ice melting may cool the lake surface. About two weeks later, lake surface temperature returned to normal status.

The spatial patterns of lake surface temperature right before (11 July) and after (19 and 27 July) the first glacier collapse (17 July 2016) are investigated by using nighttime Aqua data (Fig. 10). Lake surface temperature at the northern Aru Co was considerably lower than that at the southern Aru Co on 19 and 27 July and the lake surface temperature increased gradually from north to south, which further confirms the influence of the ice floes on lake surface temperature. This spatial pattern was in contrast with that before the glacier collapse (Fig. 10). Similar pattern also occurred in Memar Co, where lake surface temperature increased from south to north after the glacier collapse. This spatial pattern may indicate that the ice floes may flow into Memar Co through the 5 km long river (10~20 m wide) linking the two lakes.

Lake surface temperature at the southern and northern Aru Co was compared to examine its spatial heterogeneity since the northern Aru Co was closer to the two glacier collapses (Fig. 9). Before the glacier collapses (e.g., 2015), water temperature between the southern and northern Aru Co did not show considerable difference in July and August. After the glacier collapse, the lake surface temperature in August 2016 was about 1-2 $^{o}$C lower at the northern Aru Co than that at the southern Aru Co (Fig. 9). As shown in Section 4.2, most of the intruding ice into Aru Co, with a volume of $7.1 \times 10^{6}$ m$^{3}$, melted in the following two months. Since the meltwater of the intruding ice was considerably cooler than the lake water, it may decrease the lake water temperature at the northern Aru Co more significantly. This spatial difference of lake surface

temperature may indicate the impact of the meltwater on lake surface temperature in summer 2016. Similar condition can also be found in summer 2017 and 2018. Notably, the detailed process of changes in lake surface temperature after the glacier collapses is still unclear because some data are unavailable in summer due to influence of cloud cover and other factors. More work is needed to demonstrate this process by using more intensive satellite data.

>>Fig. 8<<

>>Fig. 9<<

>>Fig. 10<<

## 5 Discussion

### 5.1 Response of the rapid lake expansion on the western TP to climate change

Widespread lake expansion occurred on the interior TP during the past two decades (e.g. Lei al., 2014; Yang et al., 2017). Although there are quite a few studies reporting changes in lake area and water level on the western TP, bathymetry survey remains scarce for lakes on the western TP due to the harsh natural condition and remoteness. Qiao et al (2017) performed bathymetry survey at four lakes on the western TP, including Guozha Co, Longmu Co, Aksai Chin Lake and Bangdag Co. Their results showed that water storage at Aksai Chin Lake and Bangdag Co almost doubled during the past 40 years. At

Aksai Chin Lake and Bangdag Co, water storage increased from 1.33 to 2.57 Gt and from 1.23 to 2.60 Gt, respectively, between 1996 and 2015. In this study, our result showed that water storage at Memar Co almost doubled from 1.86 to 3.49 Gt between 1999 and 2018, which was similar to the two reported lakes. Meanwhile, based on more intense satellite data, we also found that the turning point from shrinkage to expansion at Memar Co occurred in 2000, which is about 1-2 years later than lakes in other regions of the TP (Lei et al., 2014).

Since most glaciers on the TP experienced significant mass loss during the past decades, their impact on the rapid lake expansion on the TP was investigated in many studies (Yao et al., 2010, 2018; Lei et al., 2012; Song et al., 2015; Li et al., 2017; Zhang et al., 2017; Zhou et al., 2019; Treichler et al., 2019; Brun et al., 2020). For example, glacier mass loss was estimated to contribute to ~10.5% of lake expansion at Nam Co on the central TP (Li et al., 2017). In Hol Xil region, glacier mass loss contribution to lake expansion was estimated to be 9.9 and 11.1% at LexieWudan Lake and KekeXili Lake (Zhou

et al., 2019). However, recent studies show that glaciers in the Karakoram and western Kunlun Mountains experienced balanced or slightly positive mass budgets (e.g. Kääb et al., 2015; Shean et al., 2020). Kääb et al (2018) showed that the two Aru glaciers experienced a slight mass gain of 0.2-0.3 m/yr water equivalent (w.e.) since the early 2000s, despite simultaneous glacier retreat of 520-460 m. This indicates that the glacier mass changes played a limited role in the rapid lake expansion of Memar Co since the 2000s. The melting of the two avalanches was relatively fast (3 melt seasons were

sufficient to melt the Aru-1 deposit), but even with this additional meltwater, precipitation remains the main driver of the rapid lake expansion of Memar Co.

Treichler et al (2019) suggested that the glacier thickening and rapid lake growth on the western TP were mainly attributed to the stepwise increase in precipitation since the late 1990s. Significant increase in precipitation since the 2000s is evident from meteorological station and reanalysis data (Lei et al., 2014; Treichler et al., 2019). This indicates that rapid lake expansion on the western TP, including Memar Co, was mainly a result of climate change, especially climate wetting (Lei and Yang, 2017).

## 5.2 Potential risk of lake expansion on the TP

Lake expansion on the interior TP inundated grassland and infrastructures (e.g. road and bridges) in the surrounding area, which not only led to very large economic loss, but also serious ecological and environmental problems (Yao et al., 2010; Liu et al., 2019). For example, a significant overflow suddenly occurred at Zhuonai Lake (255 km$^2$) in Hol Hil Nature Reserve in late August 2011, which further led to the overflow of Kusai Lake (260 km$^2$) and rapid expansion of the downstream lakes (Yao et al., 2012b; Hwang et al., 2019; Li et al., 2019), and even had serious consequences for antelope survival (Pei et al., 2019).

The continuous lake expansion of Memar Co may further lead to its coalescence with Aru Co in near future, which will have significant impact on the regional geomorphology and ecosystem. In 2003, the water level of Aru Co (4936.8 m a.s.l) was about 14 m higher than Memar Co (4923.2 m a.s.l), as indicated by ICESat satellite altimetry data. In 2014, CryoSat-2 data show that the elevation difference between the two lakes decreased to ~8 m due to continual lake expansion of Memar Co. After the glacier collapses, Memar Co expanded at an accelerated rate and the elevation difference became even smaller. In October 2019, the surface elevation of Memar Co reached 4931.3 m a.s.l and the elevation difference between the two lakes decreased to only 5.5 m. Projection in the near future of the increasing rate of 0.5-0.8 m/yr between 2003 and 2019, the water level of Memar Co could reach that of Aru Co in 7-11 years. According to the reconstructed relationship between lake area and lake level in Section 4.5, when the lake level of Memar Co increases by 5 m, the lake area and water storage will increase by 10.6% and 0.65 Gt, relative to 2019. At present, Memar Co is a saline lake while Aru Co is a freshwater lake. If the two lakes merge, lake salinity and ion composition will exchange freely. Memar Co will be diluted while Aru Co will be significantly salted. The habitat of the phytoplankton and zooplankton in the lakes will also change significantly in response to changes in lake salinity and ion composition. Therefore, it is necessary to carry out comprehensive monitoring at Aru Co and Memar Co in the coming years, including lake hydrology, meteorology, water quality and ecology, etc.

## 6 Conclusions

The deposit of the Aru glacier collapses on 17 July and 21 September 2016 had almost melted by September 2019 (~30% of ice remained for Aru-2 ice avalanche). A comprehensive investigation of impact of the glacier collapses on the two downstream lakes, the outflow lake Aru Co and the terminal lake Memar Co, was conducted since 2016, including

meteorology, ice mass balance, lake bathymetry and lake level changes. Based on in-situ observation and satellite data, the response of the two downstream lakes to the ice avalanches in the subsequent years (2016-2019) has been evaluated in this study. We found that the ice avalanches significantly affected the two downstream lakes during the period 2016-2019.

During Aru-1 glacier collapse, the fragmented ice mass slid into Aru Co along with a large amount of debris, which generated great impact waves in Aru Co and modified the shoreline and bathymetry. The Aru Co shoreline was pushed offshore by 100-120 m along the two sides of the first glacier collapse fan. Lake bathymetry near Aru-1 ice avalanche became much uneven, in contrast with the smooth lake floor in the adjacent areas. The intruding ice into Aru Co, with an area of ~0.89 km$^2$ and a volume of at least $7.1 \times 10^6$ m$^3$, melted in less than two months.

The spread of ice floes over the surface of Aru Co considerably lowered the lake surface temperature by 2-4 $^\circ$C in the first two weeks after the Aru-1 glacier collapse. A similar lake surface temperature decrease was also observed at Memar Co (the downsteam lake of Aru Co), but with a smaller magnitude and shorter duration. The lake surface temperature at the northern Aru Co was considerably lower than that at the southern Aru Co in summer 2016, which is mainly associated with the avalanche melting.

After the glacier collapses (2016-2019), Memar Co expanded more rapidly than before (2003-2015) as a result of faster lake level increases in summer. Between 2016 and 2019, the ice avalanche melting contributed about 23.3% of the increase in lake storage at Memar Co. If Memar Co continues to expand steadily, it will coalesce with Aru Co in 7-11 years, which could have significant impact on the regional geomorphology and ecosystem. This study also suggests the necessity for more comprehensive monitoring at Aru Co and Memar Co as significant changes may occur at the two lakes in the near future.

**Author contribution**

Lei Y., Yao T., Tian L., and Sheng Y. conceived and designed the experiments; Lei Y.B., Yao T., Tian L., Zhao H., Yang W., Zhu M., and Wu G. performed the fieldwork; Lei.Y., Yao T., Tian L., Sheng Y., Lazhu, and Yang K. analyzed the data; Liao J., Berthier E., Brun F. and Gao Y. processed the satellite data; All the authors wrote the paper.

**Competing interests**

The authors declare that they have no conflict of interest.

**Acknowledgements**

This research has been supported by the Strategic Priority Research Program of Chinese Academy of Sciences (XDA2006020102), the Second Tibetan Plateau Scientific Expedition and Research Program (2019QZKK0201), the National Natural Science Foundation of China (41971097 and 21661132003), and Youth Innovation Promotion Association

CAS (2017099). E. Berthier acknowledges support from the French Space Agency (CNES), through its TOSCA and ISIS programs. We are grateful to all the members who took part in the fieldwork.

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

**Table 1: Ice avalanche melting and its contribution to lake level increase at Memar Co between 2016 and 2019**

| Year/month | Aru-1 | | | Aru-2 | | | Total meltwater *** ($10^6$ m³) | Contribution to lake level (%) |
|---|---|---|---|---|---|---|---|---|
| | Area* (km²) | Thinning (m) | Volume loss** ($10^6$ m³) | Area* (km²) | Thinning (m) | Volume loss** ($10^6$ m³) | | |
| 2016/07-2016/10 | 9.3→8.6 | 2.8±0.1 | 32.5±1.9 | | | | 29.2±1.8 | 40.0±1.0 |
| 2016/10-2017/10 | 8.6→0 | 6.6±0.2 | 35.5±2.1 | 6.5→4.8 | 5.5±0.2 | 31.2±1.9 | 60.0±3.6 | 32.2±2.1 |
| 2017/10-2018/10 | | | | 4.8→3.0 | 5.5±0.2 | 21.3±1.3 | 19.2±1.2 | 13.6±0.7 |
| 2018/10-2019/10 | | | | 3.0→1.9 | 5.5±0.2 | 13.3±0.8 | 12.0±0.7 | 9.9±0.4 |

* For each period, the arrow separates the areas of ice deposits at the start ($S_1$) and end ($S_2$) of the period.

** The ice volume loss is calculated as circular truncated cone: $V = \frac{1}{3} \times (S_1 + S_2 + \sqrt{S_1 \times S_2}) \times dh$, where $dh$ is the reduction in ice thickness.

*** Total meltwater is derived from the volume loss of ice deposit by assuming an ice density of 900 kg/m³.

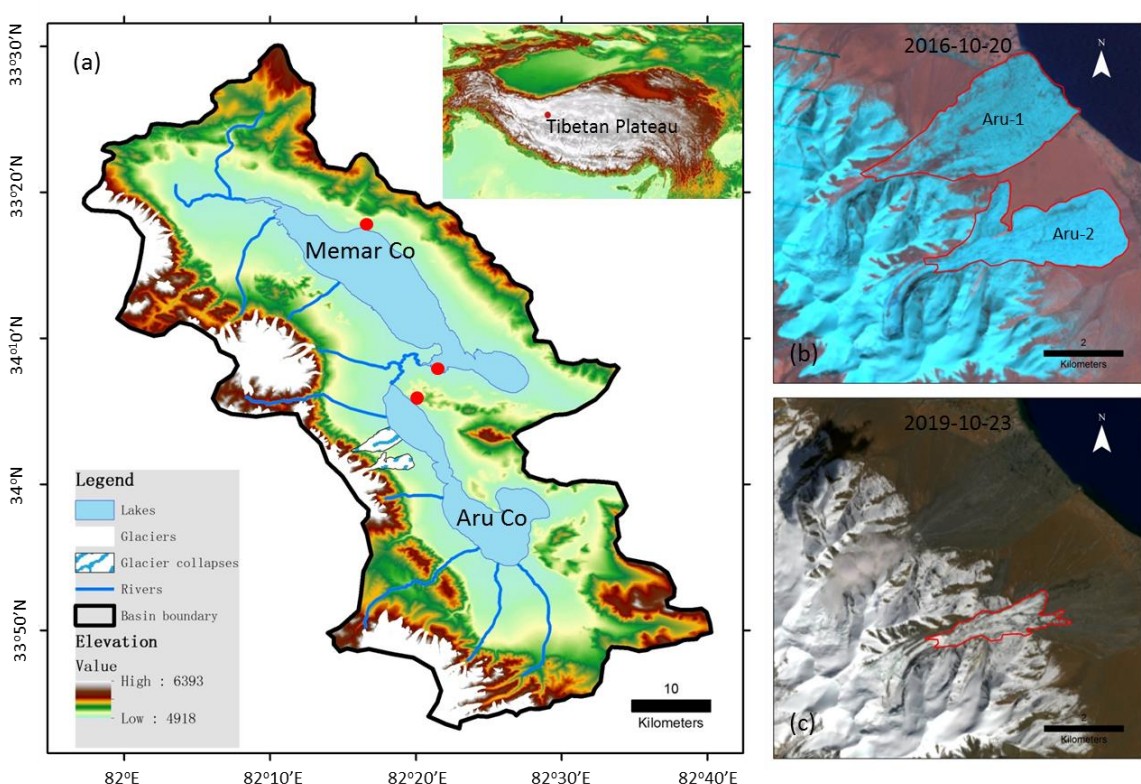

Figure 1: General description of the study area and two glacier collapses. (a): Location and general description of the study area. (b-c): Landsat satellite images of the two glacier collapses on 20 October, 2016 and 23 October, 2019. The red dots (a) denote the locations of lake level monitoring at Aru Co and Memar Co.

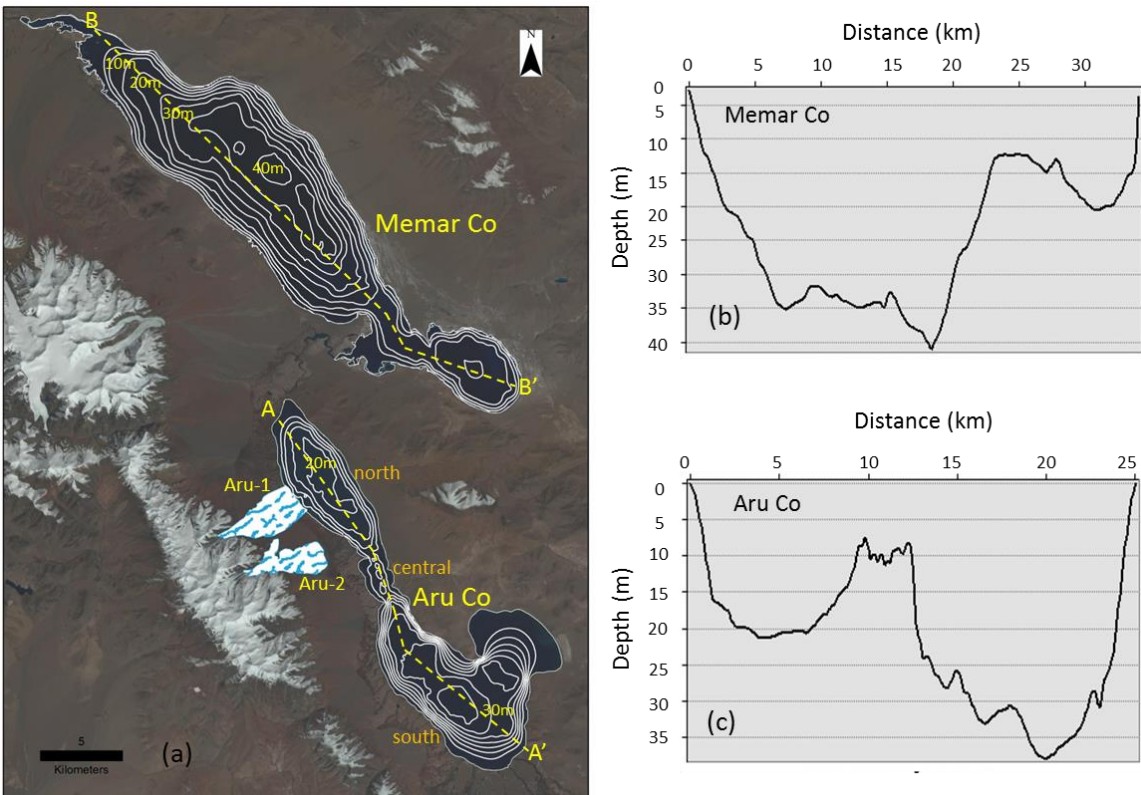


**Figure 2: The 5 m interval isobaths (a) and the water depth profiles (b, c) on NW-SE direction (the yellow dashed lines) at Aru Co and Memar Co. Landsat satellite image (a) is used to indicate the location of the lakes and glacier collapses.**



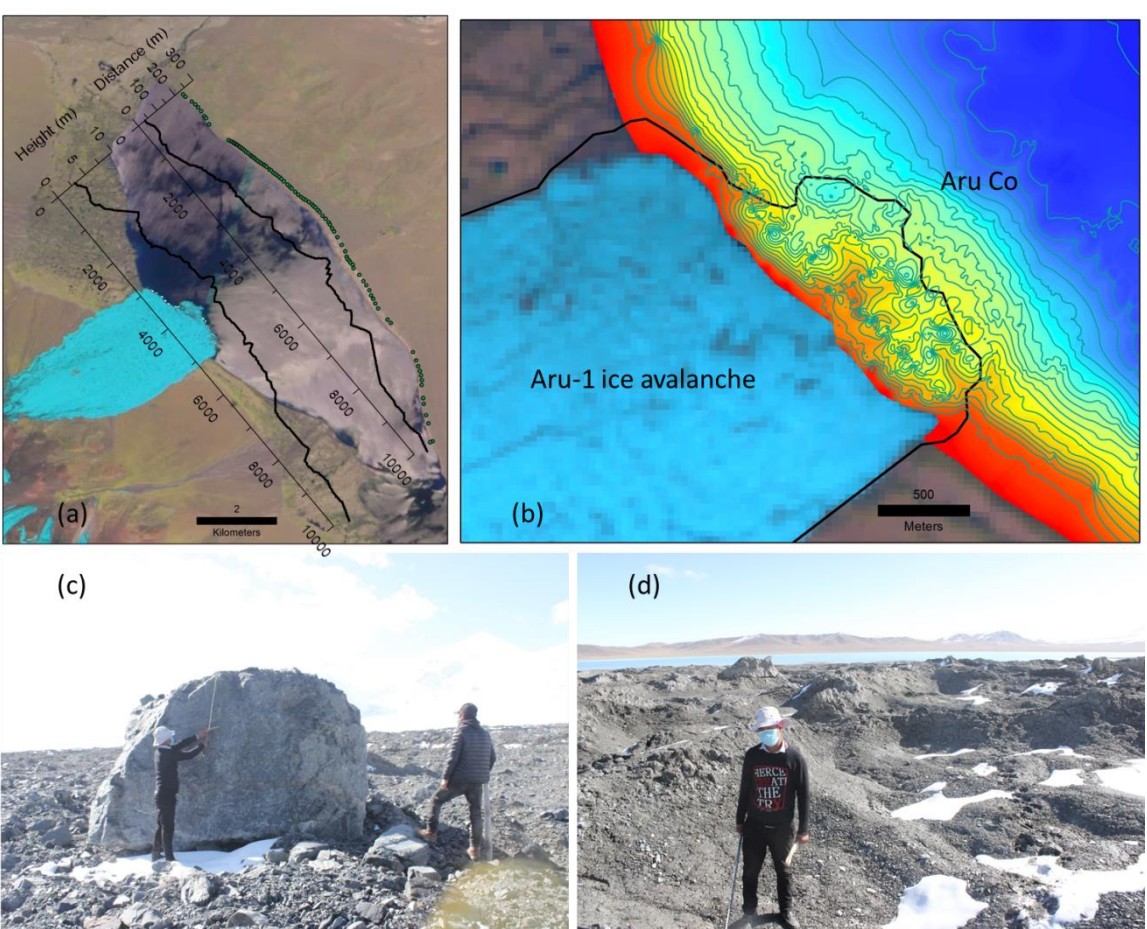

**Figure 3: The impact of Aru-1 glacier collapse on the morphology of Aru Co. a: The extent of the Aru-1 ice avalanche (Sentinel-2 image on 21 July 2016) and the impact wave at the opposite shore of Aru Co (green dots). The two curves describe the footprint of the impact wave along the eastern shoreline of Aru Co, including the distance to shoreline and the relative height to Aru Co's surface. b: The uneven lake bathymetry at Aru Co near the Aru-1 ice avalanche. The black line indicates the extent of the intruding ice into Aru Co on 21 July 2016. c, d: A large amount of debris left after the fragmented ice mass melting (photos taken on 3 October, 2019 by Yanbin Lei).**

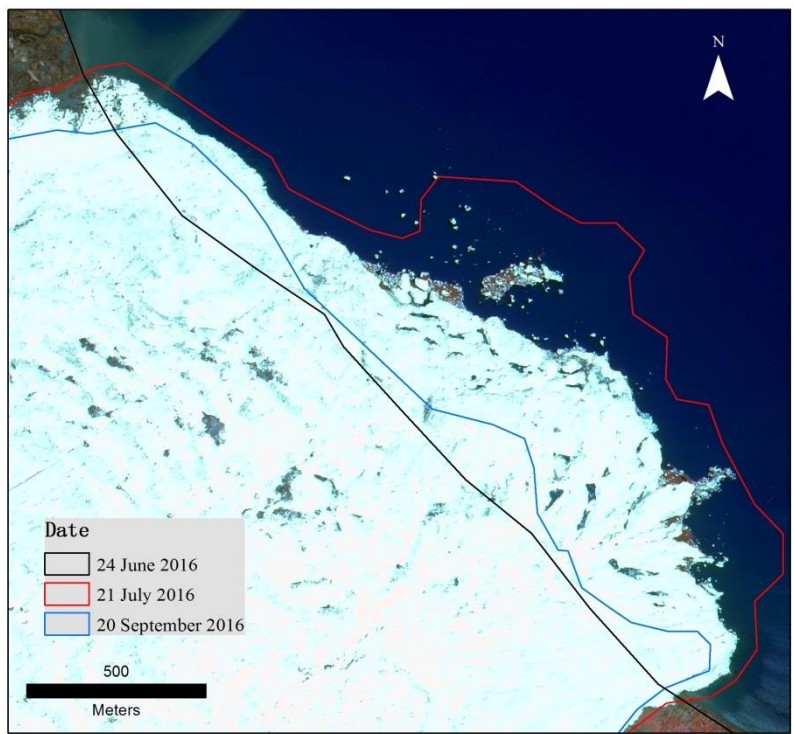

**Figure 4: The extent of the intruding ice into Aru Co Co. Gaofen-2 satellite image (1 m resolution) is used to indicate the extent of the intruding ice into Aru Co and the floating ice over the lake surface on 25 July 2016.**


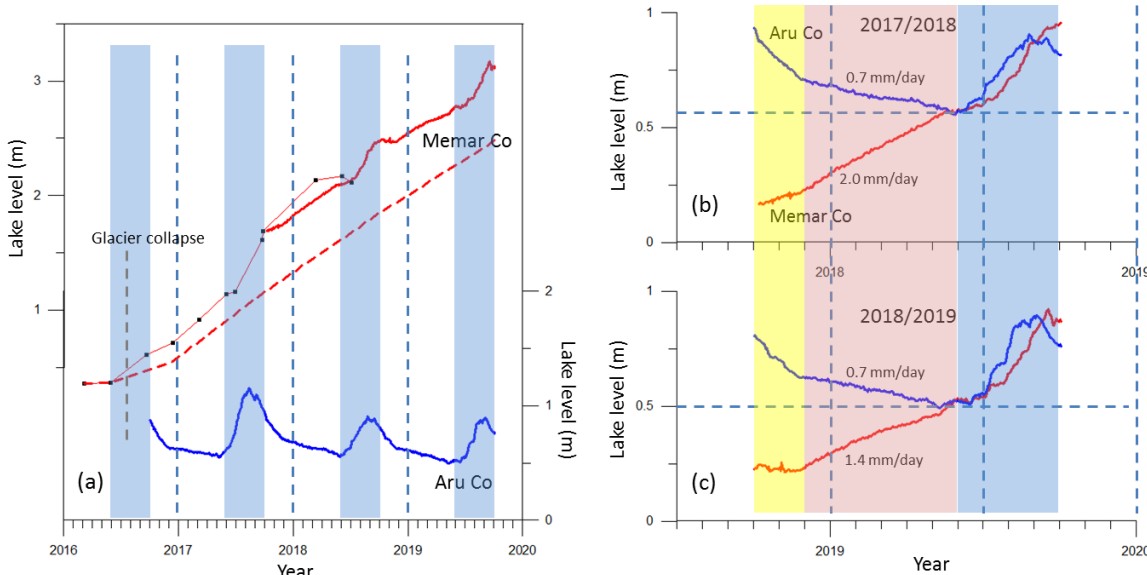


**Figure 5: In-situ lake level observations at Aru Co (blue line) and Memar Co (red line) between 2016 and 2019. (a): Lake level changes at Aru Co (blue line) and Memar Co (red line) between 2016 and 2019. The dashed red line indicates lake level changes at Memar Co without the fragmented ice melting. The black dots represents lake level derived from CryoSat-2 altimetry data. (b-c): Comparisons of lake level changes at the two lakes in 2017/2018 and 2018/2019. The coloured strips in b and c indicate different**

**periods of lake level changes in a year, namely post monsoon season, ice covered season and monsoon season.**



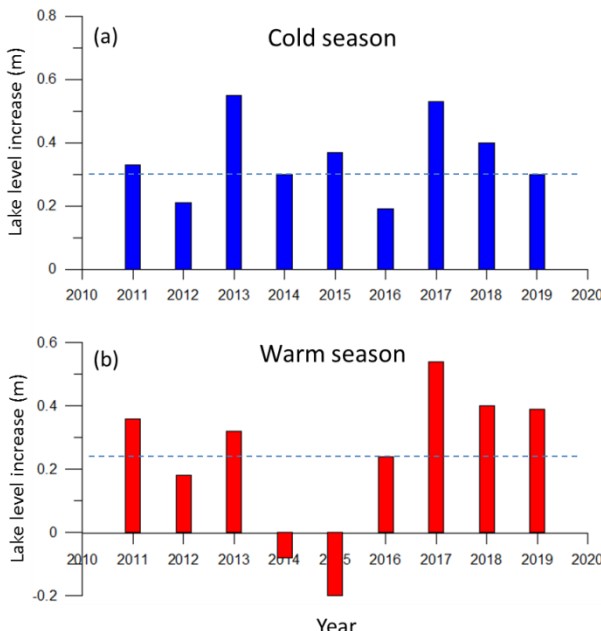


**Figure 6: Seasonal lake level changes at Memar Co derived from CryoSat-2 satellite altimetry data between 2011 and 2019. (a): Cold season (Nov to Jun). (b): Warm season (Jun to Oct).**


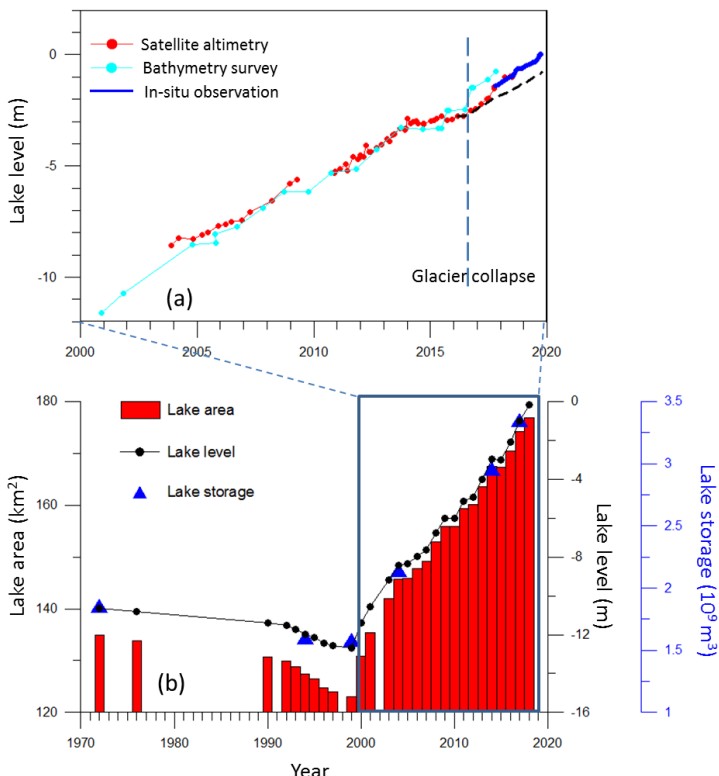

**Figure 7: Lake dynamics of Memar Co between 1976 and 2018. (a): A comparison of reconstructed lake level changes in this study (blue cycles) with satellite altimetry data (red cycles). (b): Long term changes in lake area, water level and water storage of Memar Co between 1972 and 2018. The dashed line in (a) indicates lake level changes without the fragmented ice melting.**

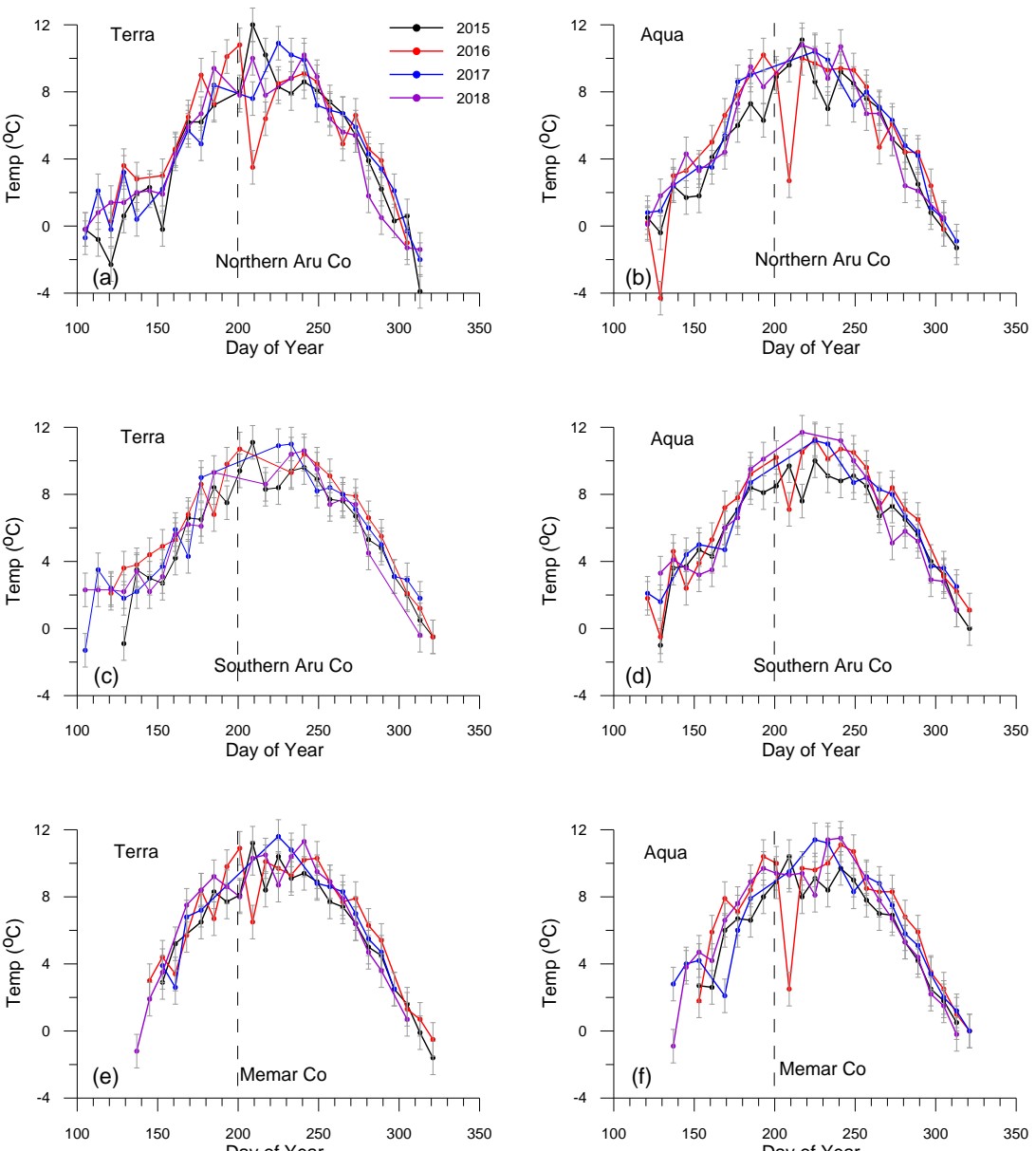

**Figure 8: Seasonal changes of nighttime lake surface temperature (LST) derived from Terra-MOD11A2 and Aqua-MYD11A2 at northern Aru Co (a-b), southern Aru Co (c-d) and Memar Co (e-f) between 2015 and 2018. The dashed lines indicate the time of the Aru-1 glacier collapse. The error bar is denoted in grey lines.**

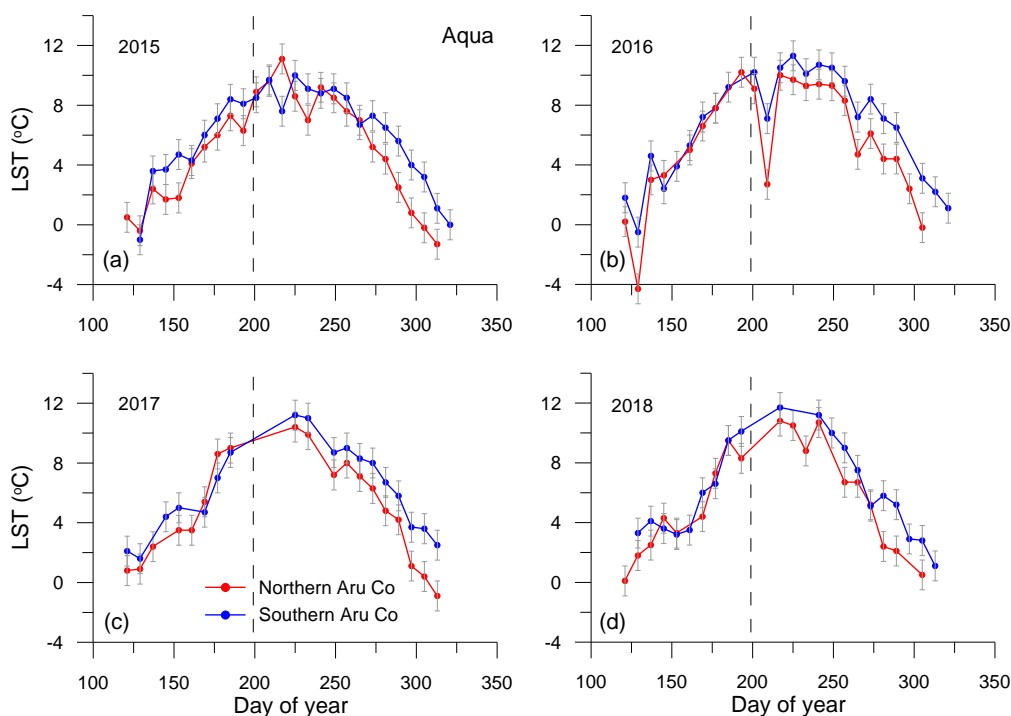

**Figure 9: A comparison of MODIS derived nighttime lake surface temperature (LST) at the northern and southern Aru Co during the period of 2016-2019. The dashed lines indicate the time of the Aru-1 glacier collapse. The error bar is denoted in grey lines.**

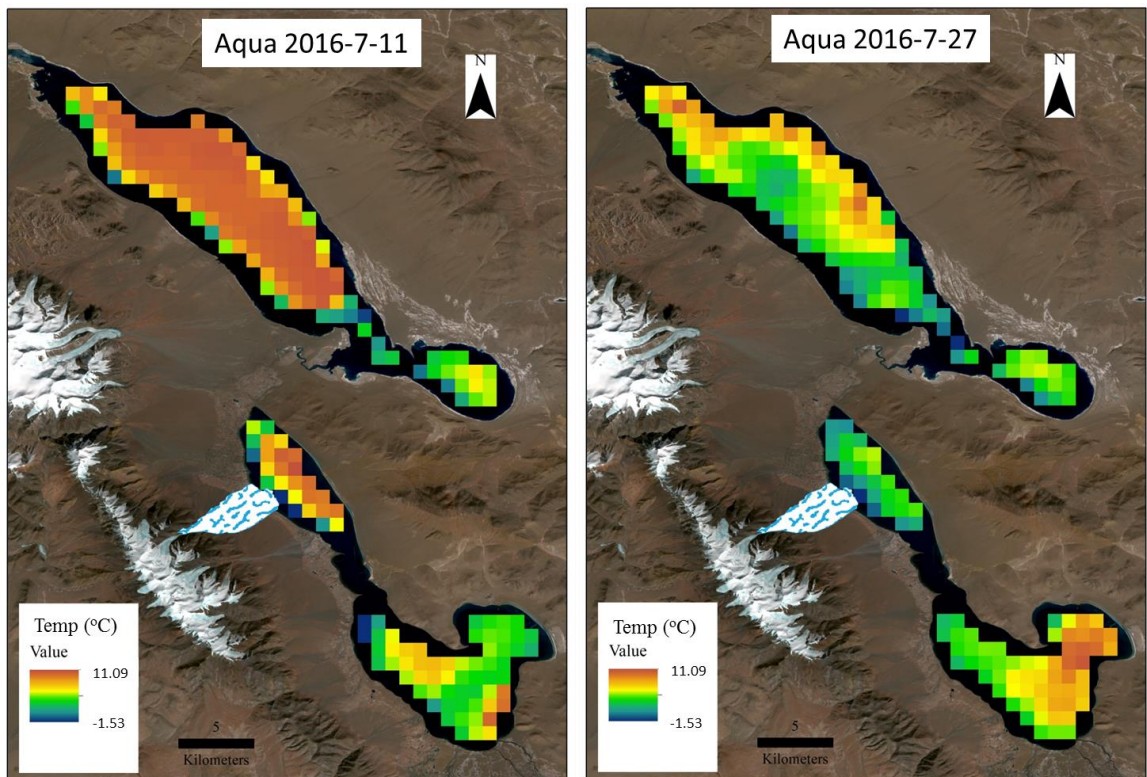

**Figure 10: Spatial distribution of nighttime lake surface temperature at Aru Co and Memar Co before (11 July 2016) and after (27 July 2016) the Aru-1 glacier collapse. Landsat satellite image is used to indicate the location of the two lakes and the Aru-1 ice avalanche.**