# Peer review of "Response of downstream lakes to Aru glacier collapses on the western Tibetan Plateau"

_The Cryosphere, 2020_

## Referee Comment (RC1) · Anonymous Referee #1 · 24 Jul 2020

The study investigates the consequences of the Aru, Tibet, glacier collapses of 2016 on the lake level, lake shore and lake surface temperatures of two nearby lakes. The study provides a number of interesting results that make it certainly worth of being published. In its present form the paper lacks however clarity in language, structure and explanations, which make it difficult to follow the findings presented. The purpose of the study should be explained better and the results presented accordingly. As now, for some of the results it is unclear how they tie into the investigation of the collapse consequences.

I recommend that at least the senior co-authors carefully revise the manuscript to make it clearer. This recommendation refers not only to language editing, but more important to the explanations given, precise language usage, and logical structure of presentation

of results. At the present state I hesitate to make large amounts of detail comments as I believe those senior co-authors should be able to see the deficiencies. Instead I give only some examples.

The paper lacks a discussion section and some discussions seem to be part of the results section. The authors should clearly separate results and their discussion/interpretation. Uncertainties in the results are hardly mentioned.

The abstract and intro most urgently need revision of language. As an example (line 39), not the Aru glaciers are giant, but their collapses! Professional language editing will likely not capture such errors. Another example, the authors say the shoreline was pushed. Did the avalanche really move the shoreline? Or did the shoreline change due to deposition of sediments? Or (line 340), does "rapid lake expansion of 0.8m/yr" refer to the lake level increase or lateral expansion of lake area? Another example for lack of clarity: in line 48 the authors talk about lake increase due to glacier melt. A few lines later (53) they write about drastic precipitation changes as cause behind lake growth.

Section 3.4: To my best knowledge, the most extensive study on lake volume changes in Tibet is Treichler et al. 2018 (https://tc.copernicus.org/articles/13/2977/2019/). The authors could compare their findings for Memar Co to the regional aggregations by Treichler et al.

Section 3.5: Any uncertainties behind the MODIS temperatures? For instance bias from undetected clouds, or lake ice?

At line 161 the lake seasonality after 2016 is presented, but it would be important to relate that to seasonality before the collapses. This is then touched upon much later.

At several occasions the authors classify the changes as "drastic" or "dramatic", for instance the 2-week lake surface cooling by 2-4 deg (line 289). Why is such change, or the other changes dramatic?

Fig 3: what is the meaning of the colored areas in panels b and c?

[Figure]

The lines in Figs 7 and 8 are difficult to compare. Better have the lines for each year combined in one plot per area? I.e. not separate plots per year but per area.

---

## Short Comment (SC1) · 26 Jul 2020

General comments

The manuscript "Tracking the impacts of the Aru glaciers collapses on downstream lakes" authored by Lei et al. presents a comprehensive investigation of the giant glacier collapse and its impact on lake shore morphology and bathymetry, lake water budgeting, and lake surface temperature (LST) of downstream lakes. The sudden, giant collapse of the Aru glaciers has attracted wide attention from the scientific and public community (Gilbert et al., 2018; Kääb et al., 2018), while its impact remains largely unresolved. This study, therefore, presents a significant and interesting story of what happened after the collapse, by combining field survey (bathymetry, LST), in-situ hy-

drological and climatological observations, and multi-source satellite observations. The results reveal that downstream lakes experienced significant changes in bathymetry, decreases in surface temperature, and accelerated expansion. Furthermore, quantitative estimation on the contribution of glacier collapse to lake expansion (26.4%) is provided by analyzing the inter-annual changes of lake water storage. The quantitative evaluation is important as that quantitative causal analysis of the lake change on the central TP remains inadequate in current literature. Whereas it has been extensively reported that most lakes on the central TP showed rapid rises in water level due to warming and wetting climate in recent decades (Song et al., 2013; Yao et al., 2018; Zhang et al., 2017). This work is beneficial to advance our understanding of the environmental response of high-altitude lakes to the effect of the changing climate and abrupt cryospheric incidence. Besides, the in-situ bathymetry data provide valuable basis for estimating the water volume of the two lakes and for continued monitoring of water storage changes.

Specific comments

Although the manuscript is generally clear and well-organized, here I raise a few issues in hoping to improve the clarity and to target the research objective better. (1) The organization of the Results part should be adjusted to focus on the evaluation of the glacier collapse influences. In Section 4.1, the description of Aru Co, Memar Co, and their hydrological connection can be moved to the part of the Study area. (2) In Section 4.4, the impact of glacier collapses and meltwater on surface temperature of two downstream lakes were analyzed. From the LST time series, it can be clearly observed that several degrees of temperature difference occurred before and after the collapse. It can be inferred that the LST differences may be revealed in the spatial pattern of MODIS-derived temperature image varying with the distance from the ice mass input place. It is thus suggested to add the maps showing the spatial pattern of LST effect responding to the glacier collapse. (3) The estimation of the collapsed glacier contribution on the lake water storage increase assumes that all of the collapsed ice mass

eventually entered the downstream lakes in the form of meltwater supply. However, the glacier melting in other forms, e.g., evaporation, may need to be discussed.

References:

Gilbert, A., Leinss, S., Kargel, J., Kääb, A., Yao, T., Gascoin, S., Leonard, G., Berthier, E. and Karki, A.: Mechanisms leading to the 2016 giant twin glacier collapses, Aru range, Tibet, Cryosphere Discuss, 1 26, doi:10.5194/tc-2018-45, 2018.

Kääb, A., Leinss, S., Gilbert, A., Bühler, Y., Gascoin, S., Evans, S. G., Bartelt, P., Berthier, E., Brun, F., Chao, W.-A., Farinotti, D., Gimbert, F., Guo, W., Huggel, C., Kargel, J. S., Leonard, G. J., Tian, L., Treichler, D. and Yao, T.: Massive collapse of two glaciers in western Tibet in 2016 after surge-like instability, Nature Geoscience, 11(2), 114–120, doi:10.1038/s41561-017-0039-7, 2018.

Song, C., Huang, B. and Ke, L.: Modeling and analysis of lake water storage changes on the Tibetan Plateau using multi-mission satellite data, Remote Sensing of Environment, 135, 25–35, doi:10.1016/j.rse.2013.03.013, 2013.

Yao, F., Wang, J., Yang, K., Wang, C., Walter, B. A. and Crétaux, J.-F.: Lake storage variation on the endorheic Tibetan Plateau and its attribution to climate change since the new millennium, Environ Res Lett, 13(6), 064011, doi:10.1088/1748-9326/aab5d3, 2018.

Zhang, G., Yao, T., Shum, C. K., Yi, S., Yang, K., Xie, H., Feng, W., Bolch, T., Wang, L., Behrangi, A., Zhang, H., Wang, W., Xiang, Y. and Yu, J.: Lake volume and groundwater storage variations in Tibetan Plateau's endorheic basin, Geophys Res Lett, 44(11), 5550–5560, doi:10.1002/2017gl073773, 2017.
* * *

---

## Referee Comment (RC2) · Anonymous Referee #2 · 30 Jul 2020

General comments The manuscript "Tracking the impacts of the Aru glaciers collapses on downstream lakes" by Lei et al. study two lakes shore morphology and bathymetry, lake water level, and lake surface temperature (LST), based on 3-4 years observation of the two collapse glaciers and two lakes downstream in TP. This study provides detail results for the important parameters of glacier lakes responding to glaciers collapse and climate change. This work is useful for understanding the relationship between glacier collapse and lakes behavior. Consequently, it is significant of environment change after the two collapses by using field observation and multi-source satellite observation. The manuscript is generally well-organized and good written. ïĄň The purpose of the study is more like two downstream lakes observation after Aru glacier collapses events. Hence, I would suggest change the title as"How two downstream lakes responding to

[Figure]

Aru glacier collapses and their changes based on in-situ and Remote sensing data " or others. ïĄň From the abstract, I got the information that the glacier collapses have two impacts on two lakes, that is, short-term (LST and lake level) and long-term impacts (Lake level and others). So, I would suggest authors refine the rules and results.

Specific comments Line 80 Aru co is . . .ïijŇhere I would suggest add a sentence " Memar co and Aru Co are lagoons" then, "Aru co is . . .." Line125 here, authors should give the methods how to get lake level changes and how to calculate the uncertainty of lake level changes. Line 130 The important feature of 2 degree decrease after collapse was success to be caught by using MODIS 8-days. And I also understood that it may be difficult to express the temperature field due to resolution (1km). But it is useful to compare between the records from AWS during Oct 2016 and Sep 2019 and LST. Line 145 here, Authors can mark where is norther basin, south basin and center part of Aru Co/Memar Co in figure 1. Line 175 did you want to express that the water level of Aru Co was controlled by climate change and the water level of Memar Co was controlled by climate change in summer and ground water in winter? Line 180 did you want to express that the Aru co has a hydraulic connection with Memar Co. And the time lag was about half an month? Line 191 Sential 2->sentinel 2 Line 208 section 4.3 this lake level and lake expansion are chaotic. It should be clear. Line 230 "In 2016" could be omitted. Line 261. I agree on your opinion that after collapse, the lake level increase in warm season rapidly. Did you have any evidence from glacier ablation observationsïij§ Line 270 the lake skin temperature? Water body temperature? Freeze up-?ice on is "Break up" melt on or melted?

---

## Referee Comment (RC3) · Anonymous Referee #3 · 7 Aug 2020

This study titled "Tracking the impacts of the Aru glacier collapses on downstream lakes" by Lei et al. is very interesting and is a good fit for the journal. This study investigated the long-term dynamics of two lakes located in the Western Tibetan Plateau in terms of physical characteristics, i.e. lake area, level, and volume using long records of remote sensing data and short in-situ records as well as field surveyed data. Moreover, this study specifically studied the consequences of glacier collapses in the catchment, which provides some implications for similar situations in the Tibetan Plateau. Although it is a very specific study, the results are worth publishing. However, there are a few concerns to be addressed before consideration for publication.

General comments: After reading the manuscript, I feel that the title is a bit too specific and does not contain what has been done in this work. I suggest rephrasing the title.

[Figure]

The hydrological connection is very interesting in my point of view. However, the reasoning of the buffering effect of the Aru Co on the Memar Co is not very convincing. L175, "discharge from Aru Co only accounted for 20-30% of the lake volume increase at Memar Co in the cold season". How is this conclusion made? Simply assume that the decline in water level completely attributes to outflow? From Lei et al. (2019 GRL), it seems the seasonality of ∼0.5 m is reasonable for endorheic lakes in the same region. It could be also possible for the Aru Co presenting a 0.5 m annual fluctuation without outflow. Outflow may happen in summer when the recharge is larger. But in cold season, whether outflow happens is questionable. It simply depends on the elevations of the Aru Co and the channel connecting the two lakes. So it needs to be careful when calculating the contribution of outflow of the Aru Co to the rising of the Memar Co by simply comparing the decline of the Aru Co and rising of the Memar Co. Another concern is the altimetry data processing, which affects the reconstruction of historical lake levels. Current methodological description is very vague. What are the data sources? How is the water level generated? How is the bias between the two data sets handled? The results relating elevation changes are heavily dependent on the bias of the two data sets.

Specific comments: L21: "collapsed suddenly" suddenly is not necessary, I think. L52: "dramatic increase", I do not think there is a dramatic increase in precipitation. Before 2014, the increasing of precipitation is not significant, and a plethora of studies debated the reason of lake expansion. Until recent years, the increasing of precipitation is much clear but not dramatic. L65-69: Do you think the bathymetry have significant change? L90: How was the snow measured? L177-178: This sentence is not clear to me. Please rephrase it. L191: "Sential" -> "Sentinel", please also change it in the caption of Figure 4. L192: Figure 3a should be Figure 4a. L209-214: How many pairs of level and area are used to build this regression model? Extrapolation based on data of six years could be problematic. This needs to be better explained. L217-218: It seems that the satellite data did not capture the sudden rise (pink dotted line) revealed in Figure 5b. Is the pink coded line indicating the reconstruction? L256-257: The seasonality revealed

by satellite data is not very clear due to the course temporal resolution.

Conclusion: I would suggest the authors try to concise the conclusions, right now too many repetitive statements from the results.

---

## Short Comment (SC2) · 10 Aug 2020

Although the manuscript is generally clear and well-organized, here I raise a few issues in hoping to improve the clarity and to target the research objective better.

Response: Thanks very much for the constructive comments and suggestions. We will carefully revise the structure of manuscript according to the comments.

(1) The organization of the Results part should be adjusted to focus on the evaluation of the glacier collapse influences. In Section 4.1, the description of Aru Co, Memar Co, and their hydrological connection can be moved to the part of the Study area.

Response: Thanks for the good suggestion. We will re-organize the structure of the paper. Result part will mainly focus on the evaluation of the glacier collapse influences.

Lake bathymetry and water storage at the two lakes will be moved to study area section. The impact of the meltwater on the seasonal lake level changes of Memar Co will be discussed in section 4.3. Lake level seasonality and the hydraulic connection will be moved to this part.

(2) In Section 4.4, the impact of glacier collapses and meltwater on surface temperature of two downstream lakes were analyzed. From the LST time series, it can be clearly observed that several degrees of temperature difference occurred before and after the collapse. It can be inferred that the LST differences may be revealed in the spatial pattern of MODIS-derived temperature image varying with the distance from the ice mass input place. It is thus suggested to add the maps showing the spatial pattern of LST effect responding to the glacier collapse.

Response: Thanks for the suggestion. We will try add a now map to show the spatial pattern of LST at Aru Co. Aru Co is narrow and long with a length of 27 km and a width of 1.4 to 9 km. To reduce the contamination from land pixels, only lake pixels beyond 1 km from shoreline were extracted. Therefore, there are limited pixels at the surface of Aru Co and there is no valid data in the central part of the lake of Aru Co because it is very narrow. Because the two ice avalanches were closer to the northern Aru Co, lake surface temperature at the southern half and northern half of the lake was extracted to investigate its spatial difference.

(3) The estimation of the collapsed glacier contribution on the lake water storage increase assumes that all of the collapsed ice mass eventually entered the downstream lakes in the form of meltwater supply. However, the glacier melting in other forms, e.g., evaporation, may need to be discussed.

Response: Thanks for the suggestion. In this study, we assume all the meltwater from the collapsed glaciers entered the downstream lakes. We do not consider evaporation or other kinds of water loss are significant because the two glacier collapses are very close to Aru Co and water loss due to evaporation should be very small and negligible.

---

## Author Comment (AC1) · 10 Aug 2020

In its present form the paper lacks however clarity in language, structure and explanations, which make it difficult to follow the findings presented. The purpose of the study should be explained better and the results presented accordingly. As now, for some of the results it is unclear how they tie into the investigation of the collapse consequences. I recommend that at least the senior co-authors carefully revise the manuscript to make it clearer. This recommendation refers not only to language editing, but more important to the explanations given, precise language usage, and logical structure of presentation of results.

Response: Thank you very much for the constructive comments and suggestions. The
language, structure and explanations have been carefully revised according to these comments. About the structure of the paper, we will make substantial revisions as following: 1, Lake bathymetry and water storage at the two lakes will be moved to study area section. Lake level changes at the two lakes will be moved to new section 4.3. 2, Section 4.1 will mainly focus on the 'The instantaneous impact of Aru-1 glacier collapse on the morphology of Aru Co'. 3, Add a new section 4.2, which is about the meltwater estimation of the two ice avalanches. The meltwater will be estimated by degree-day model according to the meteorological data and in situ measurement of glacier mass balance. 3, The impact of the meltwater on the seasonal lake level changes of Memar Co will be discussed in section 4.3. Lake level seasonality and the hydraulic connection will be moved to this part. 4, Add a discussion section, which will discuss attribution of the rapid lake expansion on the western TP and the potential risk of natural hazard on the TP.

About the purpose of the study, we will address it in more detail in the introduction. Although the mechanism of Aru glacier collapses has been investigated, its impact on the downstream lakes in the following years is still not investigated until now. Based on comprehensive observation, we will address it impact on the downstream lakes after the ice mass has melted in the following years. This study will be helpful for understand the relationship between glacier mass loss and lake behaviour in the Third Pole region under a warming climate.

About the explanation of the result, we will give more in-depth analysis. For example, we add a new Fig. 5, which is about the dynamics of the intruding ice into Aru Co. GF-2 satellite image (1 m resolution) is used to detect the floating ice over the lake surface. The extent of the intruding ice into Aru Co in summer 2016 is shown.

The paper lacks a discussion section and some discussions seem to be part of the results section. The authors should clearly separate results and their discussion/interpretation. Uncertainties in the results are hardly mentioned.

Response: A discussion section will be added in the revision. Lake level changes at Memar Co and Aru Co in the discussion part are moved to result section. In the discussion part, we mainly discuss the attribution of the rapid lake expansion on the western TP and the potential risk of natural hazard on the TP'. Uncertainties of lake level changes, water storage and lake surface temperature are evaluated in the revision.

The abstract and intro most urgently need revision of language. As an example (line 39), not the Aru glaciers are giant, but their collapses! Professional language editing will likely not capture such errors. Another example, the authors say the shoreline was pushed. Did the avalanche really move the shoreline? Or did the shoreline change due to deposition of sediments? Or (line 340), does "rapid lake expansion of 0.8m/yr" refer to the lake level increase or lateral expansion of lake area? Another example for lack of clarity: in line 48 the authors talk about lake increase due to glacier melt. A few lines later (53) they write about drastic precipitation changes as cause behind lake growth.

Response: Thanks for pointing out these errors. We will carefully revise the abstract and make it more accurate. The third paragraph in the introduction section will be moved to discussion section.

Section 3.4: To my best knowledge, the most extensive study on lake volume changes in Tibet is Treichler et al. 2018 (https://tc.copernicus.org/articles/13/2977/2019/). The authors could compare their findings for Memar Co to the regional aggregations by Treichler et al.

Response: Thanks for the good suggestions. In the discussion section, we will read this paper and discuss glacier-lake interaction at Memar Co basin in the background of the whole Tibetan Plateau, especially the western TP.

Section 3.5: Any uncertainties behind the MODIS temperatures? For instance bias from undetected clouds, or lake ice?

Response: We agree that MODIS derived lake surface temperature is easily affected by clouds and other factors, especially in summer. We will evaluate the uncertainties of MODIS derived temperature in the study method section.

At line 161 the lake seasonality after 2016 is presented, but it would be important to relate that to seasonality before the collapses. This is then touched upon much later.

Response: We add a new section about impact of glacier collapses on lake level seasonality. Lake level seasonality before and after the glacier collapse will be compared according to Cryosat-2 satellite data and in situ measurement.

At several occasions the authors classify the changes as "drastic" or "dramatic", for instance the 2-week lake surface cooling by 2-4 deg (line 289). Why is such change, or the other changes dramatic?

Response: If the changes are abrupt or larger than normal status, we classify it as dramatic. For example, the lake surface cooling by 2-4 deg in a short period is abrupt and much larger than normal status.

Fig 3: what is the meaning of the colored areas in panels b and c?

Response: The different colored areas indicate different periods of lake level changes. The three colors indicate monsoon season, post monsoon season and ice covered season. We will address this in the caption of the figure.

The lines in Figs 7 and 8 are difficult to compare. Better have the lines for each year combined in one plot per area? I.e. not separate plots per year but per area.

Response: Thanks again. We have changed this figure according to the suggestion.

---

## Author Comment (AC2) · 10 Aug 2020

General response: Thank you very much for the constructive comments and suggestions. We will revise the manuscript carefully according to these comments.

The purpose of the study is more like two downstream lakes observation after Aru glacier collapses events. Hence, I would suggest change the title as "How two downstream lakes responding to Aru glacier collapses and their changes based on in-situ and Remote sensing data " or others.

Response: Thanks for the good suggestion. Following your suggestion, we would like to change the title as 'How did the two downstream lakes respond to Aru glacier collapses?'

[Figure]

From the abstract, I got the information that the glacier collapses have two impacts on two lakes, that is, short-term (LST and lake level) and long-term impacts (Lake level and others). So, I would suggest authors refine the rules and results.

Response: Thanks for the suggestion. We will revise the abstract carefully according to this time line.

Line125 here, authors should give the methods how to get lake level changes and how to calculate the uncertainty of lake level changes.

Response: We will address this in more detail. The reconstruction of past lake level changes and its uncertainty will be addressed and estimated.

Line 130 The important feature of 2 degree decrease after collapse was success to be caught by using MODIS 8-days. And I also understood that it may be difficult to express the temperature field due to resolution (1km). But it is useful to compare between the records from AWS during Oct 2016 and Sep 2019 and LST.

Response: Thanks for the good suggestion. We agree that it may be difficult to express the temperature field due to its resolution and shape of Aru Co because the lake is very narrow and long. We will compare MODIS LST at Aru Co with air temperature from AWS between Oct 2016 and Sep 2019 in the revision.

Line 145 here, Authors can mark where is norther basin, south basin and center part of Aru Co/Memar Co in figure 1.

Response: Thanks for the good suggestion. We will show this in Figure 1.

Line 208 section 4.3 this lake level and lake expansion are chaotic. It should be clear.

Response: Thanks for the suggestion. The section 4.3 will be divided into three sections, including Section 4.2, The meltwater estimation of the two ice avalanches; Section 4.3, impact of the meltwater on the seasonal lake level changes of Memar Co; Section 4.4 impact of the meltwater on the inter-annual lake level changes of Memar

[Figure]

Co.

Line 261. I agree on your opinion that after collapse, the lake level increase in warm season rapidly. Did you have any evidence from glacier ablation observations

Response: Meltwater will also be calculated by degree-day model according to in situ observation and AWS meteorological data nearby. So the contribution of meltwater to seasonal lake level change will be further quantified.

Line 270 the lake skin temperature? Water body temperature? Freeze up-?ice on is "Break up" melt on or melted?

Response: Lake skin temperature derived from MODIS data is usually considered to be different from lake body temperature. Lake skin temperature is the water temperature at the few millimeters water depth while water body temperature is measured at the depth 30-50 cm. Freeze up is lake surface is covered by ice and break up is lake ice melts.

---

## Author Comment (AC3) · 12 Aug 2020

General comments: After reading the manuscript, I feel that the title is a bit too specific and does not contain what has been done in this work. I suggest rephrasing the title.

Response: Thank you very much for the constructive comments and suggestions. We will revise the manuscript carefully according to these comments. About the title, we will change it as 'How did the two downstream lakes respond to Aru glacier collapses?'

The hydrological connection is very interesting in my point of view. However, the reasoning of the buffering effect of the Aru Co on the Memar Co is not very convincing. L175, "discharge from Aru Co only accounted for 20-30% of the lake volume increase at Memar Co in the cold season". How is this conclusion made? Simply assume that

Interactive
comment

the decline in water level completely attributes to outflow? From Lei et al. (2019 GRL), it seems the seasonality of 0.5 m is reasonable for endorheic lakes in the same region. It could be also possible for the Aru Co presenting a 0.5 m annual fluctuation without outflow. Outflow may happen in summer when the recharge is larger. But in cold season, whether outflow happens is questionable. It simply depends on the elevations of the Aru Co and the channel connecting the two lakes. So it needs to be careful when calculating the contribution of outflow of the Aru Co to the rising of the Memar Co by simply comparing the decline of the Aru Co and rising of the Memar Co.

Response: Thanks for raising this question. The hydraulic connection between Aru Co and Memar Co is not the main topic of this study, so we do not want to address this in more detail. The lake level increase in cold season at Memar Co is very interesting and we will analyze this special lake level seasonality in another paper. Yes, the percentage of 20-30% is simply calculated according to lake volume changes of Aru Co and Memar Co during ice covered season. As we have shown in the text, the two lakes are covered by lake ice between December and May. During this cold period, lake level of Aru Co decreased slightly while Memar Co increased dramatically. The decrease in lake storage at Aru Co only accounted for 20-30% of the lake volume increase at Memar Co, so we believe that the lake surplus at Memar Co is not mainly supplied by discharge of Aru Co, but by other source of water discharge, for example groundwater supply. It is true that the seasonal lake level fluctuation is in a range of 0.5 m and we agree that it is questionable to compare the decline of the Aru Co and rising of the Memar Co when the lake does not freeze up.

Another concern is the altimetry data processing, which affects the reconstruction of historical lake levels. Current methodological description is very vague. What are the data sources? How is the water level generated? How is the bias between the two data sets handled? The results relating elevation changes are heavily dependent on the bias of the two data sets.

Response: Thanks for the good suggestions. We will address the processing of altimetry data in more detail in the revision. ICESat altimetry data, which was processed according to Li et al (2014), was used to examine water level variations between 2003 and 2009. CryoSat-2 data, which was processed according to Xue et al (2018), were used to examine water level variations between 2010 and 2018. Notably, the two datasets are referenced to different ellipsoids and geoid height. The ICESat data contains corrected surface ellipsoidal heights referenced to TOPEX/Poseidon ellipsoid and geoid height referenced to Earth Gravity Model (EGM) 2008; while the CryoSat-2 data are referenced to WGS84 and EGM96 (Song et al., 2015). In order to make the two datasets comparable, lake elevation at Aru Co is compared because the lake is an outflow lake and inter-annual lake level changes are relatively small. As shown in Fig. 5, lake level at Aru Co was in its lowest in May and it is very stable from year to year determined by the outlet. The ICESat derived lake surface elevations of Aru Co were averaged to be 4936.67 m a.s.l. in April during the period 2003-2009 (n=2, Stdev=0.01). The CryoSat-2 derived lake surface elevations of Aru Co were averaged to be 4937.04 m a.s.l. in May during the period 2011-2016 (n=5, Stdev=0.08). This elevation difference of 0.37 m is considered to be the bias of the two datasets at the study area.

Specific comments:

L52: "dramatic increase", I do not think there is a dramatic increase in precipitation. Before 2014, the increasing of precipitation is not significant, and a plethora of studies debated the reason of lake expansion. Until recent years, the increasing of precipitation is much clear but not dramatic.

Response: This part will be moved to discussion part because another reviewer suggests that it is not closely related to the subject of the study. Precipitation exhibited dramatic inter-annual fluctuations on the interior TP, so the dramatic increase was not obvious at some stations. However, the overall increase in precipitation is very clear in most stations of the interior TP.

L65-69: Do you think the bathymetry have significant change?

Response: The ice avalanches influence lake bathymetry mainly near the collapse fan, not the whole the lake (section 4.1).

L90: How was the snow measured?

Response: The snow was measured by a T200B rain gauge.

L191: "Sential" -> "Sentinel", please also change it in the caption of Figure 4.

Response: Thanks for pointing out this error.

L192: Figure 3a should be Figure 4a.

Response: Thanks for pointing out this error.

L209-214: How many pairs of level and area are used to build this regression model? Extrapolation based on data of six years could be problematic. This needs to be better explained.

Response: In this study, six pairs of lake level and area are used. Since these data contains the lowest (∼1997) and highest level, the regression model used in this study is reliable.

L217-218: It seems that the satellite data did not capture the sudden rise (pink dotted line) revealed in Figure 5b. Is the pink coded line indicating the reconstruction?

Response: The pink dote line is the satellite altimetry data. The dramatic increase of lake level change occurred during the period between 2016 and 2019. To be honest, the sudden rise in lake level at Memar Co after the Aru-1 collapse can not be captured due to its temporal resolution.

L256-257: The seasonality revealed by satellite data is not very clear due to the course temporal resolution.

Response: We agree with this. Because the lake was also rapidly expanding before the glacier collapse, the lake level seasonality revealed by Cryosat-2 data did not exhibit big

difference before and after the collapse. However, if we compare the average values during the two periods, we can find the considerable difference of lake level change in summer.

Conclusion: I would suggest the authors try to concise the conclusions, right now too many repetitive statements from the results.

Response: Thanks, we will rephrase the conclusion carefully.
* * *

---

## Author Response (AR1)

**Reply to reviewer #1**

In its present form the paper lacks however clarity in language, structure and explanations, which make it difficult to follow the findings presented. The purpose of the study should be explained better and the results presented accordingly. As now, for some of the results it is unclear how they tie into the investigation of the collapse consequences. I recommend that at least the senior co-authors carefully revise the manuscript to make it clearer. This recommendation refers not only to language editing, but more important to the explanations given, precise language usage, and logical structure of presentation of results.

Reply: Thank you very much for the constructive comments and suggestions. The language, structure and explanations have been carefully revised according to these comments.

For the structure of the paper, we have made substantial revisions as following:

1, Add a new Section 4.3, which mainly focuses on the meltwater estimation of the two ice avalanches. The meltwater is mainly estimated by area of the ice mass and in-situ measurement of glacier mass balance (Tab. 1). This estimation is further validated by elevation changes of the two ice avalanches (Section 4.3).

2, Add a new Section 4.4, which mainly focuses on the impact of the meltwater on the seasonal lake level changes at Memar Co. Lake level seasonality and the hydraulic connection are moved to this part.

3, Add a discussion section (Section 5), which focuses on the response of the rapid lake expansion on the western TP to climate change and the potential risk of natural hazard on the TP.

For the purpose of the study, we have addressed it in more detail in the introduction (line 47-59). Although the mechanism of Aru glacier collapses has been investigated, its impact on the downstream lakes in the subsequent years (2016-2019) has still not been investigated until now. Based on comprehensive in-situ observations and satellite data, we investigate its impact of the two glacier collapses on the downstream lakes in the subsequent years when most of the ice mass has melted. This study not only provides us unique evidence of the impact of a large amount of glacier melting on the downstream lakes, but also helps to improve our understanding the relationship between glacier mass loss and lake behavior on the TP under a warming climate.

For the explanation of the result, we have added further discussion in the revision. For example, we added two new figures in the revision. One (Fig. 4) is about the dynamics of the intruding ice into Aru Co. High resolution (1 m resolution) GF-2 satellite image is used to detect the extent of the intruding ice and the floating ice over the lake surface. The dynamics of the intruding ice into Aru Co in summer 2016 is shown in this figure. The other (Fig. 10) is about the spatial distribution of lake surface temperature before and after the glacier collapses.

The paper lacks a discussion section and some discussions seem to be part of the results section. The authors should clearly separate results and their discussion/interpretation.

**Uncertainties in the results are hardly mentioned.**

Reply: A discussion section (Section 5) has been added in the revision. In this new section, we mainly discuss the response of the rapid lake expansion on the western TP to climate change and the potential risk of natural hazard on the TP.

Uncertainties of lake level changes, water storage and lake surface temperature are evaluated in the revision as well (Line 104, 150, Line 161).

The abstract and intro most urgently need revision of language. As an example (line 39), not the Aru glaciers are giant, but their collapses! Professional language editing will likely not capture such errors. Another example, the authors say the shoreline was pushed. Did the avalanche really move the shoreline? Or did the shoreline change due to deposition of sediments? Or (line 340), does "rapid lake expansion of 0.8m/yr" refer to the lake level increase or lateral expansion of lake area? Another example for lack of clarity: in line 48 the authors talk about lake increase due to glacier melt. A few lines later (53) they write about drastic precipitation changes as cause behind lake growth.

Reply: Thanks for pointing out these errors. We have carefully revised the abstract and make it more accurate.

Part of the third paragraph in the introduction is moved to discussion section (Section 5.2) in the revision.

**Section 3.4: To my best knowledge, the most extensive study on lake volume changes in Tibet is Treichler et al. 2018 (https://tc.copernicus.org/articles/13/2977/2019/). The authors could compare their findings for Memar Co to the regional aggregations by Treichler et al.**

Reply: Thanks for the good suggestions. In the discussion section, we now use the main result of Treichler et al. 2018 as the background of lake expansion on the western TP and discuss glacier-lake interaction in Memar Co basin.

**Section 3.5: Any uncertainties behind the MODIS temperatures? For instance bias from undetected clouds, or lake ice?**

Reply: We agree that MODIS derived lake surface temperature is easily affected by clouds and other factors, especially in summer. We evaluate the uncertainties of MODIS derived temperature in the method section (Line 161).

**At line 161 the lake seasonality after 2016 is presented, but it would be important to relate that to seasonality before the collapses. This is then touched upon much later.**

Reply: We added a new section (Section 4.4) about the impact of glacier collapses on lake level seasonality. Lake level seasonality before (2011-2015) and after (2016-2019) the glacier collapses is compared according to Cryosat-2 satellite data and in-situ measurement.

**At several occasions the authors classify the changes as "drastic" or "dramatic", for instance the 2-week lake surface cooling by 2-4 deg (line 289). Why is such change, or the other changes dramatic?**

Reply: Thanks for the good suggestion. We agree that using 'drastic' or 'dramatic' in some places are not accurate. We have deleted or replaced some of them in the revision.

**Fig 3: what is the meaning of the colored areas in panels b and c?**

Reply: The three different colors in figure 3 indicate monsoon season, post monsoon season and ice covered season. We have addressed this now in the caption of the figure.

The lines in Figs 7 and 8 are difficult to compare. Better have the lines for each year combined in one plot per area? I.e. not separate plots per year but per area. Reply: Thanks again. We have changed this figure according to the suggestion.

**Reply to reviewer #2**

The purpose of the study is more like two downstream lakes observation after Aru glacier collapses events. Hence, I would suggest change the title as "How two downstream lakes responding to Aru glacier collapses and their changes based on in-situ and Remote sensing data" or others.

Reply: Thanks for the good suggestion. Following your suggestions, the title of the paper is revised as 'Response of downstream lakes to Aru glacier collapses on the Tibetan Plateau'.

From the abstract, I got the information that the glacier collapses have two impacts on two lakes, that is, short-term (LST and lake level) and long-term impacts (Lake level and others). So, I would suggest authors refine the rules and results.

Reply: Thanks for the suggestion. We have revised the abstract carefully according to this time line.

**Specific comments:**

*Line 80 Aru co is : : here I would suggest add a sentence "Memar co and Aru Co are lagoons" then, "Aru co is : : :."*

Reply: Thanks for the suggestion. We have revised this sentence according to this suggestion (line 78-79).

*Line125 here, authors should give the methods how to get lake level changes and how to calculate the uncertainty of lake level changes.*

Reply: We have addressed the method about lake level reconstruction in more detail in the revision (Line 143-153). The uncertainty of past lake level changes is also estimated (Line 150)

Line 130 The important feature of 2 degree decrease after collapse was success to be caught by using MODIS 8-days. And I also understood that it may be difficult to express the temperature field due to resolution (1km). But it is useful to compare between the records from AWS during Oct 2016 and Sep 2019 and LST.

Reply: Thanks for the good suggestion. We added a new Fig. 10 about the spatial distribution of lake surface temperature in the revision. We agree that it is difficult to express the temperature field because Aru Co is very narrow and long. There are no valid data in the central part of Aru Co.

We included a comparison between MODIS LST at Aru Co and air temperature from AWS in 2017 and 2018 in the revision (Fig. S5). Daily air temperature had larger fluctuation than water temperature and was always higher than lake surface temperature at Aru Co.

Fig. S5: Comparison of MODIS derived lake surface temperature with in-situ measurement at the shoreline and daily air temperature from AWS station.

*Line 145 here, Authors can mark where is norther basin, south basin and center part of Aru Co/Memar Co in figure 1.*

Reply: Thanks for the good suggestion. We have shown this in Figure 1.

Line 175 did you want to express that the water level of Aru Co was controlled by climate change and the water level of Memar Co was controlled by climate change in summer and ground water in winter?

Reply: Yes, we have addressed this more clearly in the revision (Line 266-272).

**Line 180 did you want to express that the Aru co has a hydraulic connection with Memar Co. And the time lag was about half a month?**

Reply: Yes, it should be hydraulic connection and we have revised this sentence in the revision (Line 266). From the seasonal pattern of lake level changes at the two lakes, there is about half a month lag.

**Line 191 Sential 2->sentinel 2**

Reply: Thanks for pointing out this error.

**Line 208 section 4.3 this lake level and lake expansion are chaotic. It should be clear.**

Reply: Thanks for the suggestion. The former Section 4.3 is now divided into three sections in the revision:

- Section 4.3, The meltwater estimation of the two ice avalanches;
- Section 4.4, The impact of the meltwater on the seasonal lake level changes of Memar Co;
- Section 4.5, The impact of the meltwater on the inter-annual lake level changes of Memar Co.

**Line 230 "In 2016" could be omitted.**

Reply: Thanks for pointing out this error. We have revised it (Line 223).

Line 261. I agree on your opinion that after collapse, the lake level increase in warm season

**rapidly. Did you have any evidence from glacier ablation observations**

Reply: Meltwater from the two ice avalanches is estimated according to ice avalanche area and changes in ice thickness (Section 4.3). In-situ observation of thickness change was conducted in the first two years (2016 and 2017). Meltwater from the avalanche deposits is constrained using examination of satellite images and differencing of digital elevation models (DEMs). The contribution of meltwater to seasonal lake level change is further quantified (Line 279-281).

**Line 270 the lake skin temperature? Water body temperature? Freeze up-?ice on is "Break up" melt on or melted?**

Reply: Lake skin temperature derived from MODIS data is usually considered to be different from water body temperature. Lake skin temperature is the water temperature of the uppermost 10-20  $\mu$ m deep molecular layer while water body temperature is water temperature of several cm to <1 m.

Yes, freeze up means that lake surface is covered by ice and break up means that lake ice melts.

**Reply to reviewer #3**

General comments: After reading the manuscript, I feel that the title is a bit too specific and does not contain what has been done in this work. I suggest rephrasing the title.

Reply: Thank you very much for the constructive comments and suggestions. We have revised the manuscript carefully according to these comments.

About the title, we change it as 'Response of downstream lakes to Aru glacier collapses on the Tibetan Plateau'

The hydrological connection is very interesting in my point of view. However, the reasoning of the buffering effect of the Aru Co on the Memar Co is not very convincing. L175, "discharge from Aru Co only accounted for 20-30% of the lake volume increase at Memar Co in the cold season". How is this conclusion made? Simply assume that the decline in water level completely attributes to outflow? From Lei et al. (2019 GRL), it seems the seasonality of 0.5 m is reasonable for endorheic lakes in the same region. It could be also possible for the Aru Co presenting a 0.5 m annual fluctuation without outflow. Outflow may happen in summer when the recharge is larger. But in cold season, whether outflow happens is questionable. It simply depends on the elevations of the Aru Co and the channel connecting the two lakes. So it needs to be careful when calculating the contribution of outflow of the Aru Co to the rising of the Memar Co by simply comparing the decline of the Aru Co and rising of the Memar Co.

Reply: Thanks for the comment. The hydraulic connection between the two lakes is investigated by comparing the seasonal lake level changes at Aru Co and Memar Co. 'Lake level at Aru Co started to increase rapidly in early July, which was about half a month earlier than that at Memar Co. Meanwhile, the end of the rapid lake level increase at Aru Co was also about half a month earlier than that at Memar Co. The time lag of seasonal lake level changes at the two lakes indicates the buffering effect of Aru Co as an outflow lake. A large amount of water was stored in Aru Co in summer, and released to Memar Co in autumn. In early September, lake level at Aru Co decreased by about 10 cm, accounting for about 90% of the lake volume increase at Memar Co. This indicates that Aru Co, as an outflow lake, plays a significant role in regulating the water balance of Memar Co.'

As shown in the main text (Line 251-252), the two lakes are covered by lake ice between December and May. During the ice covered period, lake level of Aru Co decreased slightly while Memar Co increased dramatically. The decrease in lake storage at Aru Co only accounted for 20-30% of the lake volume increase at Memar Co during this period, so we believe that the lake surplus at Memar Co is not mainly contributed by the discharge from Aru Co. It is true that the seasonal lake level fluctuation is in a range of 0.5 m and we agree that it is questionable to compare the decline of the Aru Co and rising of the Memar Co when the lake does not freeze up.

Another concern is the altimetry data processing, which affects the reconstruction of historical lake levels. Current methodological description is very vague. What are the data sources? How is the water level generated? How is the bias between the two data sets handled? The results relating elevation changes are heavily dependent on the bias of the two data sets.

Reply: Thanks for the good suggestion. We have addressed altimetry data processing in more detail in the revision (Line 121-132). 'ICESat altimetry data was processed after Li et al (2014) and was used to examine water level variations between 2003 and 2009. CryoSat-2 data was processed after Xue et al (2018) and was used to investigate water level variations between 2010 and 2018. Both lakes were observed by ICESat satellite twice or three times a year (Phan et al., 2012), and by CryoSat-2 satellite every two or three months (Kleinherenbrink et al., 2015; Jiang et al., 2017). Notably, the two datasets are referenced to different ellipsoids and geoid height. The ICESat data contains corrected surface ellipsoidal heights referenced to TOPEX/Poseidon ellipsoid and geoid height referenced to Earth Gravity Model (EGM) 2008; while the CryoSat-2 data are referenced to WGS84 and EGM96 (Song et al., 2015). In order to make the two datasets comparable, lake elevation at Aru Co is compared because the lake is an outflow lake and inter-annual lake level changes are relatively small. At Aru Co, the lowest lake level in May is very stable from year to year as it is controlled by the elevation of the outlet. The ICESat and CryoSat-2 derived lake surface elevations of Aru Co were averaged to be 4936.67 m a.s.l. in April (n=2) during the period 2003-2009 and 4937.04 m a.s.l. in May (n=5) during the period 2011-2016, respectively. The small elevation difference of 0.37 m is considered to be the bias of the two datasets and used to correct satellite altimetry data.'

*Specific comments: L21: "collapsed suddenly" suddenly is not necessary, I think.* Reply: We have deleted it in the revision.

**L52: "dramatic increase", I do not think there is a dramatic increase in precipitation. Before 2014, the increasing of precipitation is not significant, and a plethora of studies debated the reason of lake expansion. Until recent years, the increasing of precipitation is much clear but not dramatic.**

Reply: Thanks for the suggestion. 'Dramatic' is not accurate some places, so we replaced it with other words or deleted it in the revision.

The response of lake expansion to climate change is discussed in a new section (Section 5.1) because it is not closely related to the subject of this study. Yes, precipitation on the TP exhibited significant spatial difference and different precipitation dataset shows quite large difference. This is mainly due to lack of in situ measurement. On the interior TP, precipitation data is only available at several stations and exhibits large inter-annual fluctuations. It should be noted that lake can expand when precipitation is higher than the equilibrium value, so lake expansion does not need continual increase in precipitation. Generally, the precipitation was above average value on the interior TP after the late 1990s, so we can find that most lakes expanded rapidly during the past 20 years.

**L65-69: Do you think the bathymetry have significant change?**

Reply: The ice avalanches can influence lake bathymetry of Aru Co near the collapse fan, not the whole lake (Section 4.2).

**L90: How was the snow measured?**

Reply: The snow is measured by a T200B rain gauge (Line 87).

**L177-178: This sentence is not clear to me. Please rephrase it.**

Reply: Thanks, we have rephrased it in the revision (Line 264-265).

*L191: "Sential" -> "Sentinel", please also change it in the caption of Figure 4.* Reply: Thanks for pointing out this error. We have revised it in the revision.

**L192: Figure 3a should be Figure 4a.**

Reply: Thanks for pointing out this. We have rephrased this sentence in the revision.

**L209-214: How many pairs of level and area are used to build this regression model? Extrapolation based on data of six years could be problematic. This needs to be better explained.**

Reply: In this study, six pairs of lake level and area are used, including 1972, 1994, 1999, 2004, 2014 and 2018. Since these data contains the lowest (~1997) and highest (2018) lake area and water level, we believed the regression model used in this study is reliable.

**L217-218: It seems that the satellite data did not capture the sudden rise (pink dotted line) revealed in Figure 5b. Is the pink coded line indicating the reconstruction?**

Reply: The pink dotted line is the satellite altimetry data. The dramatic increase of lake level change occurred during the whole period between 2016 and 2019. To be honest, the sudden rise in lake level at Memar Co shortly after the Aru-1 collapse can not be captured by CryoSat satellite data due to its temporal resolution.

**L256-257: The seasonality revealed by satellite data is not very clear due to the course temporal resolution.**

Reply: We agree with this. Because Memar Co also expanded rapidly before the glacier collapse, the lake level seasonality revealed by Cryosat-2 data did not exhibit big difference before and after the collapse. However, if we compare the average values between the two periods, we can find the considerable difference of lake level change in summer.

**Conclusion: I would suggest the authors try to concise the conclusions, right now too many repetitive statements from the results.**

Reply: Thanks, we have rephrased the conclusion carefully.

**Reply to Short comments**

(1) The organization of the Results part should be adjusted to focus on the evaluation of the glacier collapse influences. In Section 4.1, the description of Aru Co, Memar Co, and their hydrological connection can be moved to the part of the Study area.

Reply: Thank you very much for the constructive comments and suggestions. We re-organize the structure of the paper in the revision. Lake level seasonality and the hydraulic connection are moved to section 4.4, which is about the impact of the meltwater on the seasonal lake level changes of Memar Co. We do not move lake bathymetry and water storage at the two lakes to study area section because they belong to part of the result in this study. If we move them to the study area, readers may have question about how these results come from.

(2) In Section 4.4, the impact of glacier collapses and meltwater on surface temperature of two downstream lakes were analyzed. From the LST time series, it can be clearly observed that several degrees of temperature difference occurred before and after the collapse. It can be inferred that the LST differences may be revealed in the spatial pattern of MODIS-derived temperature image varying with the distance from the ice mass input place. It is thus suggested to add the maps showing the spatial pattern of LST effect responding to the glacier collapse.

Response: Thanks for the good suggestion. We add a new figure 10 in the revision about the spatial pattern of lake surface temperature (LST). The spatial patterns of LST before (11 July) and after (19 and 27 July) the first glacier collapse are investigated in Section 4.6. Before the first glacier collapse, the spatial pattern of lake surface temperature on 11 July 2016 is investigated based on MYD11A2 data. After the first glacier collapse, the spatial patterns of lake surface temperature on July 19th and 27th, 2016 are investigated. Because Aru Co is narrow (1.4 to 9 km) and only lake pixels beyond 1 km from shoreline were extracted, there was no valid data in the central part of Aru Co.

The spatial pattern of LST shows that the northern Aru Co was considerably cooler than the southern Aru Co after the glacier collapse (19 and 27 July 2016), which is in contrast with that before the glacier collapse (11 July 2016). This is because the ice avalanche was closer to the northern Aru Co. Similar pattern also occurred in Memar Co, where lake surface temperature increased from south to north. This spatial pattern may also indicate that the floating ice from the first ice avalanche also influenced the lake surface temperature of Memar Co through the 5 km long river (10~20 m wide) linking the two lakes.

**(3) The estimation of the collapsed glacier contribution on the lake water storage increase assumes that all of the collapsed ice mass eventually entered the downstream lakes in the form of meltwater supply. However, the glacier melting in other forms, e.g., evaporation, may need to be discussed.**

Reply: Thanks for the suggestion. In this study, we assume all the meltwater from the collapsed glaciers entered the downstream lakes. According to in-situ observation by Li et al. (2019), sublimation and/or evaporation at Guliya ice cap on the western TP were estimated to be 0.12 m in the year 2015/2016. Sublimation and evaporation is relatively small and negligible compared with the rapid melting of the avalanche deposit. Meanwhile, the two

glacier collapses are very close to Aru Co. Therefore, we do not consider evaporation or other kinds of water loss in this study (Line 227-228).

Li, S., Yao, T., Yu, W., Yang, W., Zhu, M.: Energy and mass balance characteristics of the Guliya ice cap in the West Kunlun Mountains, Tibetan Plateau. Cold Reg. Sci. Technol., 159, 71–85, 2019.

**Tracking the impacts of the Aru glacier collapses on downstream** lakes**

**Response of two-downstream lakes to Aru glacier collapses on the Tibetan Plateau**

- Yanbin Lei1,2\*, Tandong Yao1,2, Lide Tian2,3, Yongwei Sheng4, Lazhu5, Jingjuan Liao6, Huabiao Zhao1,
   2,99, Wei Yang1, 2, Kun Yang2, 7, Etienne Berthier8, Fanny Brun8, Yang Gao1,2, Meilin Zhu1, Guangjian Wu1, 2
  - 1Key Laboratory of Tibetan Environment Changes and Land Surface Processes, Institute of Tibetan Plateau Research, Chinese Academy of Sciences, Beijing 100101, China
- 2CAS Center for Excellence in Tibetan Plateau Earth System, Beijing, 100101, China
   3Institute of International Rivers and Eco-security, Yunnan University, Kunming, China
   4Department of Geography, University of California, Los Angeles (UCLA), CA 90095–1524, USA
   5National Tibetan Plateau Data Center, Institute of Tibetan Plateau Research, Chinese Academy of Sciences, Beijing 100101, China
- 6Key Laboratory of Digital Earth Science, Institute of Remote Sensing & Digital Earth, Chinese Academy of Sciences, Beijing 100094, China 7Department of Earth System Science, Tsinghua University, Beijing 10084, China
  - 8LEGOS, CNES, CNRS, IRD, UPS, Universit éde Toulouse, Toulouse, France
- 20 9Ngari-Ngari Station for Desert Environment Observation and Research, Institute of Tibetan Plateau Research, Chinese Academy of Sciences, 100101, China

Correspondence to: Yanbin Lei (leiyb@itpcas.ac.cn)

Abstract The entire lower parts of two<del>Two giant glaciers</del> glaciers (termed Aru 1 and Aru 2) at the Aru range, western Tibetan Plateau, at the Aru range on the western Tibetan Plateau (TP) collapsed collapsed suddenly unprecedentedly on 17

- 25 17-July and 21-21-September 2016, respectively, respectively, causing fatal damage to local people and their livestock. The giant ice avalanches, with a total volume of 150×106 m3, had almost melted by September 2019. How the two downstream lakes (i.e. the outflow Aru Co and the terminal Memar Co) responded to the glacier collapses is still not unclear investigated. Based on in-situ observation, bathymetry survey and satellite data, here we show the impacts of the two ice avalanchesglacier collapses on the downstream lakes, the outflow Aru Co and the terminal Memar Co, in terms of lake
- morphology, water level and water temperature in the subsequent four years (2016-2019). After the first glacier collapse, the ice avalanche slid into Aru Co along with a large amount of debris, which generated great wave at Aru Co and significantly modified the lake's shoreline and bathymetryunderwater topography. The intruding ice with a volume of at least 7.1×106 m3 soon spread over the Aru Co's surface and dramatically lowered lake surface temperature (LST) by 2-4 °C in the first 4-2 weeks after the first glacier collapse. Due to the large amount of meltwater input, By comparing with long term lake level ehangeswe found that Memar Co exhibited more rapid expansion after the glacier collapses (2016-2019) than before (2003-

2014) due to the large amount of meltwater input,, characterized by much larger ILake level increase in cold season did not exhibit considerable difference, but itbecame much larger in warm season. Assuming all the meltwater could be transferred into Memar Co. tThe melting of ice avalanches was found to contribute to about 26.4% 30%-of the increase in lake storage between 2016 and 201926.4. Out results -Assuming all the meltwater could be transferred into Memar Co. its contribution to the annual lake level increase was estimated to be 41.9% 34.3% 14.2% and 10.3%, respectively, between 2016 and 2019.

[revised manuscript text omitted]

100 terms of lake bathymetry, lake surface water temperature and lake level changes at both lakes. (The structure of the paper

**2 General description of the sStudy area**

Aru Co and Memar Co are located in an endorheic basin on the western Tibetan Plateau (Fig. 1). According to the second glacier inventory (Guo et al., 2015), 105 pieces of glaciers are located in the basin with a total area of ~184 km2. Studies showed that glaciers in this region had been rather stable in the past decades (Tian et al., 2017; Zhang et al., 2018). Two

- adjacent glaciers (Aru-1 and Aru-2) to the west of Aru Co collapsed suddenly on July 19th and September 21st, 2016, respectively, killing nine people and hundreds of livestock. The fragmented ice mass of the first-Aru-1 glacier collapse reached Aru Co at high speed after running out 6-7 km beyond the glacier terminus, generating huge impact wave at Aru Co (K ääb et al., 2018). Fieldwork aAt the firstAru-1 glacier collapse fan, showed that the depth of the collapsed fragmented ice mass varied from 3 m at the glacier snout to 13 m at the far end of the deposit (Tian et al., 2017). The two ice avalanches
- 110 covered an area of 9.4 and 6.7 km2, and their volumes of the detached glacier were estimated to be 68 and  $83 \times 10^6$  m3, respectively (Tian et al., 2017; K ääb et al., 2018).

Aru Co and Memar Co are located in the two\_downstream of lakes of the two-glacier collapses (Fig. 1). Both lakes are lagoons and share a catchment area of 2310 km2. Aru Co is an outflow lake with salinity of 0.56g/L, and Memar Co is the terminal lake of Aru Co with salinity of 6.22 g/L. The surface elevation of Aru Co (4937 a.s.l.) was about 14 m higher than

115 Memar Co (4923 a.s.l.) in 2003, according to ICESat satellite altimetry data (Li et al., 2014). There are dozens of visible paleo-shorelines around Memar Co. The highest shoreline around Memar Co is ~40 m above the modern lake level, indicating Aru Co and Memar Co used to be one large lake on a geological time scale.

are lagoons and The two lakes share a catchment area of 2310 km2. Aru Co is an outflow lake with salinity of 0.56g/L, and Memar Co with salinity of 6.22 g/L is the terminal lake of Aru Co. The surface elevation of Aru Co (4937 a.s.l.) was about

120 14 m higher than Memar Co (4923 a.s.l.) in 2003, according to ICESat satellite altimetry data (Li et al., 2014). There are dozens of visible paleo-shorelines around Memar Co. The highest shoreline around Memar Co is ~40 m above the modern
lake level, indicating Aru Co and Memar Co used to be one large lake on a geological time scale.

The climate in this area is cold and dry most of the year. Automatic weather station (AWS) data collected between Oct 2016 and Sep 2019 near the glacier collapse (~5000 a.s.l.) show that mean annual air temperature is -3.6 °C, with the lowest value

125 of -14.0 °C in January and the highest value of 7.2 °C in August. A T200B rain gauge data during the same period-indicated that mean annual precipitation near the glacier collapse is 333 mm between October 2016 and September 2019, which is much higher than that at Nagri meteorological station (Tian et al., 2017). Precipitation in this region is mainly concentrated in the warm season from June to September, accounting for more than 80% of annual precipitation. Snowfall in the cold season between October and May only accounts for 10-15% of annual precipitation.

130 >>Fig. 1
- 175
   The two lakes was observed by ICESat satellite detected the two lakes-twice or three times a year (Phan et al., 2012), and by

   CryoSat-2 satellite observed the two lakes every two or three months (Kleinherenbrink et al., 2015: Jiang et al., 2017).

   Notably, the two datasets are referenced to different ellipsoids and geoid height. The ICESat data contains corrected surface

   ellipsoidal heights referenced to TOPEX/Poseidon ellipsoid and geoid height referenced to Earth Gravity Model (EGM)

   2008; while the CryoSat-2 data are referenced to WGS84 and EGM96 (Song et al., 2015). In order to make the two datasets
- 180 comparable, lake elevation at Aru Co is compared because the lake is an outflow lake and inter-annual lake level changes are relatively small. As shown in Fig. 5At; lake level at Aru Co, was in itsthe lowest lake level in May-and it is very stable from year to year determined by the outlet. The ICESat and CryoSat-2 derived lake surface elevations of Aru Co were averaged to be 4936.67 m a.s.l. in April (n=2) during the period 2003-2009-(n=2). The CryoSat-2 derived lake surface elevations of Aru Co were averaged to be and 4937.04 m a.s.l. in May (n=5) during the period 2011-2016, respectively-(n=5). This elevation
- 185 difference of 0.37 m is considered to be the bias of the two datasets atin this study.

**3.4 Long-term lake level reconstruction**

Lake level variations before 2003 were determined based on the current water depthslake bathymetry and the position of past shorelines, which is derived from Landsat satellite images (Lei et al., 2012). The primary objective of bBathymetric survey was used to determine the current water depth over shorelines that were previously exposed (Lei et al., 2012). To minimize errors, more than 10 bathymetry lines across Memar Co were acquired and used to

- (Lei et al., 2012). To minimize errors, more than 10 bathymetry lines across Memar Co were acquired and used to reconstruct past lake level changes. In this study, IL ake level changes in 19772, 1994, 1997, 1999, 2004 and 2014 relative to October 2018 were reconstructed by bathymetry survery. We used as many asMore than 10 bathymetry lines across Memar Co were acquired and used to reconstruct past lake level changes. Memar Co exhibited shrinkage from 1972 to 1999, and then expanded significantly since 2000. Therefore, different stages of lake level changes are included in this reconstruction. Uncertainty of lake level changes is mainly determined by the resolution of satellite
- $\frac{1}{1}$

200 Continual lake level changes at Memar Co since 1972 were reconstructed using this relationship and the corresponding lake area.

**3.5 Lake surface temperature derived from MODIS satellite data**

- In this study, MODIS 8-day land surface temperature products (i.e. MOD11A2 and MYD11A2) were used to investigate changes in lake surface temperature at Aru Co and Memar Co. The MODIS 8-day data is the averaged lake surface temperature of daily MODIS product over eight days. In both platforms (Terra and Aqua), two instantaneous observations were collected every day (Terra: approximately 10:30 and 22:30 local time, Aqua: approximately 13:30 and 01:30 local time). The MODIS 8 day data is the averaged lake surface temperature of daily MODIS product over eight days. Only nighttime data was used in this study because there was less cloud cover at night (Zhang et al., 2014; Wan et al., 2018). MOD11A2 and MYD11A2 products are produced at a spatial resolution of about 1 km-with Thean accuracy of MODIS LST data is-1 K in most cases-under clear sky conditions (Wan, 2013). MODIS lake surface temperature data are-is pre-
- processed to account for atmospheric and surface emissivity effects. The cloud mask (MOD35) used for inland water provides a surface temperature measurement when there is a 66 % or greater confidence of clear-sky conditions (Wan2 2013), otherwise no temperature is produced. To reduce the contamination from land pixels, only lake pixels beyond 1 km from shoreline were extracted (Ke et al., 2014)-(Fig. S3). Because the two ice avalanches were closer to Atthe northern Aru Co (Fig. 2), lake-lake surface temperature at the southern half (29 pixels) and northern half (7 pixels) of the lake was extracted to investigate its spatial difference. At Memar Co, lake surface temperature at the northern half of the lake (81 pixels) was extracted. Anomalous lake surface temperature was examined and removed if there was big difference between the two MOD11A2 and MYD11A2 datasets. To confirm the reliability of MODIS products, nighttime lake surface temperature was
- compared with in-situ observation at the shoreline.

----Results

**225 4.1 Bathymetry survey at Aru Co and Memar Co**

Aru Co has a surface area of  $105 \text{ km}^2$  with a length of 27 km and a width of 1.4 to 9 km. The bathymetry survey shows that Aru Co is composed of two sub-basins. The northern basin accounts for less than 30% of the total lake area with a maximum water depth of 20 m. The southern basin is the main body of Aru Co, with a maximum water depth of 35 m (Fig. 2). The

central part of Aru Co is narrow and shallow, with a width of ~1.5 km and a maximum water depth of ~11 m. The entire Aru 230 Co has an average water depth of 17.6 m and total water storage of  $17.9 \times 10^8$  m3.

**>>Fig. 2**

**4.2 4.2 The instantaneous Impact impact impact of the Aru-1-first glacier collapse on the morphology of Aru Co**

Aru-1 glacier collapse ran into Aru Co at high speed after running out 6-7 km beyond the glacier terminus (Tian et al., 2017; K ääb et al., 2018). A Sentinel-2 satellite image acquired on July 21st, 2016 showed that the ice avalanche ran into Aru Co as far as ~800 m and the intruding ice into Aru Co had an area of ~0.89 km2 with a width of ~2250 m and an average length of 400 m. about 0.89 km2 of ice intruded into Aru Co. The shoreline of Aru Co was pushed eastward ~400 m on average (Fig. 3a). The intruding ice generated great wave impact at the northern Aru Co due to its high speed and large volume, which inundated the opposite shore of Aru Co (K ääb et al., 2018). Fieldwork in October 2016 showed that there was clear footprint of wave erosion at the opposite shore of the northern Aru Co, which extended up to 240 m inland and 9 m above the lake

level along 10 km long shoreline distance (Fig. 3a).

Bathymetry survey in July 2017 showed that water depth at the east margin of the intruding ice into Aru Co-was about 8 m,  $\frac{1}{2}$ . Because the intruding ice was obviously higher than the lake surface, indicating that it the water depth of 8 m was probably

- 250 the least thickness of the ice mass into the lakeAru Co-as the intruding ice are obviously higher than the lake surface. Therefore, regardless of the floating ice over the lake surface, the volume of ice mass into Aru Co is estimated to be at least  $7.1 \times 10^6$  m3, accounting for ~10% of the total ice volume of the firstAru-1 glacier collapse. Due to the influence of lake water, the intruding ice melted quickly in less than two months as indicated by Comparison with-Landsat satellite image on September 20th, 2016-on 20th September, 2016 shows that most of the ice mass into Aru Co melted in two months(Fig. 4).
- 255

260

We conducted a dDetailed bathymetry survey at Aru Co near the first glacier collapse fanshowed that the underwater topography near Aru-1 ice avalanche was largely modified.— duDue to a large amount of debris input along with the fragmented ice mass, the lake bathymetry was largely modified. Fig. 4-3b shows that the uneven-bathymetry near the ice avalancheglacier collapse fan-became uneven, which is quite different from the adjacent areas,—. The extent of the uneven lake bathymetry was slightly larger than that of the intruding ice on July 21st, 2016 (Fig. 3b), indicating that part of the intruding ice had spread over the surface of Aru Co or melted in four days after the glacier collapse. The uneven underwater

topography indicated that a large amount of debris was transported into Aru Co or the lake bed was significantly eroded which indicates that the lake bed was greatly eroded. The lake bottom stays unchanged in areas deeper than 15 m or far from the glacier collapse fan.

265

An investigation of Aru-1 the first-ice avalanche in October 2019 gave further evidence of debris input into Aru Co.

**The >>Fig. 4<<**

- Clear deposit with a thickness of 0.2-1.0 m of the first glacier collapse fan was investigated in October 2019[eft after the fragmented ice mass had completely melted. We found that tThe original road was no longer accessible because the glacier collapse fanit was covered by thick a large amount of debris-with a thickness of 0.2-1.0 m. Boulders with a diameter of 1-2 m –were found even near the lake-shoreline (Fig. 4d3d). The uneven land surface may explains well why the lake bottom became uneven. Fieldwork also showed that Due to the large amount of debris input, the Aru Co's shoreline near the northern and southern sides of the ice avalanche dramatically at the northern and southern sides of the glacier collapse fan was pushed moved eastward offshore for about 100-120 m, which was
- 275 northern and southern sides of the glacier collapse fan was pushed moved eastward offshore for about 100-120 m, which was probably due to the deposit of debris transported by glacier collapse and afterwards meltwater. This indicates that the debris of first glacier collapse significantly modified the land surface and the lake bathymetry of Aru Co.

>>Fig. 4<mark>3</mark><<

>>Fig. 4<<

280

**4.23 The meltingwater estimation of the two ice avalanchesice avalanches-(degree day model)**

According to the areas and volumes reported by Käb et al (2018), Both satellite images and fieldwork showed that the first glacier collapses have almost melted by October 2019, the average thickness of Aru-1 and Aru-2 ice avalanches was estimated to be 7.6 m and 15.2 m, respectively. Different thickness of the fragmented ice mass determined the duration of its

285 melting. The Aru-1first glacier collapse had completelyalmost melted in two summers as indicated by bSatellite imagesy October 2018 in October 2017 (only some scattered ice mass left)Fig. 1. (Supplementary). Areal changes?The melting of Aru-2 glacier collapse lasted longer due to its larger thickness. In October 2019,

Only less than 0.5 km2 of the fragmented ice hadremained an area of about 1.9 km2, accounting for about 29% of the total area. at The remaining ice mass mainly occurred in the upper part of Aru-2 second-ice avalancheglacier collapse fan, where

290 the fragmented ice iwas thicker (K ääb et al., 2018) by October 2019. Areal changes? DEM difference of ice left. Here we made a roughly estimatione the yearly meltwater of the fragmented ice mass according to the area and in-situ measurements of ice mass balance-in the first two years. In 2016, in-situ measurements at 9 sites show that Aru-1 the-ice massavalanche thinned about 2.84 m on average between August 13th 2016 and Oct 24th 2016, which corresponded to about a volume of 30.624.4×106 m3-of meltwater (assuming the ice density of 900 kg/m3). Considering the intruding ice into

|---|-------|-----|------|------|

| 295 | Aru Co $(7.1 \times 10^6 \text{ m}^3)$ , the total meltwater of the first ice avalanche is estimated to be $28.4 \times 10^6 \text{ m}^3$ in 2016 (assuming the ice     |
|-----|-------------------------------------------------------------------------------------------------------------------------------------------------------------------------|
|     | density of 900 kg/m 3 ). The meltwater of Aru-2 ice avalanche is not considered in 2016 because air temperature was already                                  |
|     | close to zero degree in the late September. The largest melting of the fragmented ice mass occurred in summer 2017                                                      |
|     | according to Landsat satellite images and in situ observation. In-situ measurements show Aru-1 and Aru-2 ice avalanches                                                 |
|     | melted down 6.5 m and 5.5 m on average, respectively, between September 2016 and September 2017. Most of the first ice                                                  |
| 300 | avalanche had melted by October 2017 and, it In situ measurements show the first and second glacier collapses melted down                                               |
|     | 6.5 m and 5.5 m on average, respectively, between September 2016 and September 2017, meltwater in 2017 is considered to                                                 |
|     | be 26.6 $\times 10^6$ m 3 , which is also the remaining part of Aru-1 ice avalanche. Meltwater of the second ice avalanche is estimated                      |
|     | to be 33.2 $\times 10^6$ m 3 . Thus, the total meltwater in 2017 is estimated to be 59.7 which corresponds to 63.9 $\times 10^6$ m 3 of meltwater |
|     | in total. ByIn October 2018 and 2019, only a small portion of the second glacier collapse remained only had an area of 3.0                                              |

305 and 1.9 km2, respectively. We assumed that the ratemeltdown of the ice meltingmass at the second glacier collapse in 2018 and 2019 wasere same as in 2017, and the total volume of meltwater is estimated to be  $25.24.0 \times 10^6$  m3, and  $18.2 \times 10^6$  m3 in 2018 and 2019, respectively (Tab. 1). Thus, about  $3.1 \times 10^6$  m3 of the fragmented ice was left at Aru-2 ice avalanche according to the above calculation. By October 2019, the second glacier collapse had also completely melted, with the remaining meltwater of  $18.2 \times 10^6$  m3 (Tab. 1).

310 >>Tab. 1<<

4.<del>3</del>4

**4.3 Impact of the jee avalanchesmeltwater on the seasonal lake level changes of Memar Co of Memar Co**

5 The impact of the twomeltwater ice avalanches on seasonal and inter annual lake level changes of mainly occurred at Memar Co was investigated because their meltwater finally went into Memar Co via Aru Co is an outflow lake. Here we

[revised manuscript text omitted]

accelerated speed after the Aru glacier collapses in 2016. Between 2003 and 2014, the lake level of Memar Co increased

|---|-------------------------------------------------|
(宋体),加粗,(中文)中文(中国) |
(宋体),加粗,(中文)中文(中国) |
(宋体),(中文)中文(中国)    |

- 360 steadily at a rate of 0.59 m/yr. The lake expansion paused in 2015, in response to the widespread drought over the TP during the strong 2015/2016 El Niño event (Lei et al., 2019). Between 2016 and 2019After the first glacier collapse, the lake level of Memar Co expanded increased more rapidly with an average rate of 0.80 m/yr-between 2016 and 2019, which was about 30% higher than that between 2003 and 2014. The lake level and the water storage and storage at of Memar Co accumulatively increased by 3.0 m and and 0.38 Gt, respectively, the tween 2016 and 2019. Assuming all the
- meltwater can be transferred into Memar Co, the total melting of ice avalanches contributed to 26.4% of increase in lake
   storage between 2016 and 2019. We can see that without the melting of ice avalanches, the rate of lake level increase at of
   Memar Co after the glacier collapses could be similar to that between 2003 and 2014 (Fig. 3a7a).
   The contribution of the ice avalanches melting on inter-annual lake level changes of Memar Co is further-also quantitatively

evaluated. In 2016, when ice melting mainly occurred in the first glacier collapse, Memar Co expanded slightly with lake
 level increase of 0.43 m. In 2017, when the ice melting reached its peak, Memar Co exhibited the most dramatic expansion,
 with lake level increase of 1.07 m. In 2018 and 2019, when the ice melting slowed down, the Memar Co expanded expansion
 of Memar Co also slightlyslowed down, with lake level increase of 0.8 m and 0.69 m, respectively. Assuming all the
 meltwater could be transferred into Memar Co, its contribution to the lake level increase of Memar Co is estimated to be

41.938.8%, 3432.31%, 1417.20% and 1015.30% of the total lake level increase in the subsequent 4 years (2016, 2017, 2018) and \_2019), respectively.

>>Fig. 7
| 415 | (e.g., 2015), water temperature between the southern and northern Aru Co did not exhibit considerable difference in July and                      |                    |  |
|     | August (Fig. 7a). After the glacier collapse, the lake surface temperature in August 2016 at the northern Aru Co-was about 1-                     |                    |  |
|     | 2 °C lower at the northern Aru Co in August 2016 than that at the southern Aru Co (Fig. 79b). We attribute tThis spatial                          | 带格式的: 上标           |  |
|     | difference temperature difference tomay indicate the long term impact of the meltingwater on lake surface temperature in                          |                    |  |
|     | summerof the intruding ice. Satellite imagesAs showedn in section 4.1, most of that-the intruding ice into Aru Co, with a                         |                    |  |
| 420 | volume of 7.1×10 6 m 3 , melted by September 20th, 2016 in two months (two months after the first glacier collapses). Since | 带格式的: 上标           |  |
|     | the meltwater of the intruding ice of the intruding ice-was considerably coolder than the lake water, the meltingit of the                        | 带格式的: 上标           |  |
|     | intruding ice-may cooldecrease the lake water temperature at the northern Aru Co more significantly. Similar condition can                        |                    |  |
|     | also be found in summer 2017 and 2018.                                                                                                            |                    |  |
|     | Although MODIS derived lake surface temperature can be affected by cloud cover and other factors (Ke et al., 2014), both                          |                    |  |
| 425 | confirmedimpact of ice avalanches on MOD11A2 and MYD1A2 products recorded a dramatic decrease of lake surface                                     |                    |  |

|     | because there is no data available due to influence of cloud cover and other factors. We attribute the dramatic decrease in lake |                       |
|-----|----------------------------------------------------------------------------------------------------------------------------------|-----------------------|
|     | surface temperature to the floating ice over the surface of Aru Co. Since the meltwater of the intruding ice was considerably    |                       |
|     | colder than the lake water. The dramatic decrease in lake surface temperature at Aru Co indicates that although the volume       |                       |
| 430 | of the ice avalanches only account for a small portion of lake water storage at Aru Co (less than 8%), its melting could have    |                       |
|     | dramatic impact on lake surface temperature. Notably, M More work is still-needed to demonstrate this process the detailed       |                       |
|     | process of changes in lake surface temperature after the glacier collapsesby using more intensive satellite data.                |                       |
|     |                                                                                                                                  |                       |
|     | >>Fig. 78<<                                                                                                   |                       |
| 435 | >>Fig. <del>89</del><<                                                                                        |                       |
|     | >>Fig. 10<<                                                                                                   |                       |
|     |                                                                                                                                  |                       |
|     | 5 Discussion                                                                                                                     | (禾体), (甲义) 甲义(甲国)     |
|     | 5 Discussion                                                                                                              |                       |
|     | 5.21 Attribution of Demonses of the world lake amongion on the western TD to alimete shonge                                      |
|     | 5.21 Attribution of Response of the rapid take expansion on the western 1 r to chinate change                                    |                       |
| 440 | Widespread lake expansion occurred on the interior TP during the past two decades (e.g. Lei al., 2014). Although there are       |                       |
|     | many studies about changes in lake area and water level, Most endorheielson the western TP significantly dedsince the late       |                       |
|     | 1990s. Lei et al (2014) showed that the total area of 10 large lakes on the western TP increased by 18.2% between 1976 and       |                       |
|     | 2010. The lake level increased at an average rate of 0.3 m/yr according to ICESat satellite altimetry data between 2003 and      |
|     | 2008. The extent of lake area and water level increase on the wester TP is similar with lakes in other regions of the TP. Yao    |                       |
| 445 | showed that the total water storage at ?? lakes increased at a rate of (??) between 2003 and 2008. However, changes in total     |                       |
|     | lake volumeBbathymetry survey areis still less investigated conducted at lakes on the western TP due to its harsh natural        |                       |
|     | condition and remoteness. Qiao et al (20197) conducted bathymetry survey at four lakes on the western TP, including              |
|     | Guozha Co, Longmu Co, Aksai Chin Lake and Bangdag CoTheir results showed that lake volumewater storage at Aksai                  |                       |
|     | Chin Lake and Bangdag Co was almost doubled during the past 40 years. At Aksai Chin Lake and Bangdag Co, water                   |                       |
| 450 | storage of at-increased from 1.3283 to 2.5687 Gt and from 1.23 to 2.60 Gt, respectively, frombetween 1996 toand 2015. At         |                       |
|     | Bangdag Co, wincreased from 1.226 to 2.598 Gt during the same period. In this study, our result showed that water storage        |                       |
|     | at Memar Co increased from 1.9958 to 3.49 Gt at Memar Co between 197799 and 2018, which was similar with the two                 |                       |
|     | reported lakes. Meanwhile, Bbased on more intense inter annual lake level changessatellite data (Fig. ?), we also foundalso      |                       |
|     | found that the turning point from shrinkage to expansion at Memar Co occurred at 2000, which is about 1-2 years later than       |                       |
| 455 | lakes in other regions of the TP (Lei et al., 2014).                                                                             |                       |
|     | Since most glaciers are widely distributed on the TP experienced dramatic mass loss during the past decades, its impact on       |                       |
|     | the rapid lake expansion on the TP is often connected with regional glacier meltingwas investigated in many studies (e.g.        |                       |

Yao et al., 2010, 2018; Lei et al., 2012; Song et al., 2015; Li et al., 2017; Zhang et al., 2017; Zhou et al., 2019; Treichler et al., 2019). For example, glacier mass loss was estimated to contribute to ~10.5% of lake expansion at Nam Co on the central TP (Li et al., 2017). In Hol Xil region, glacier mass loss contribution to lake expansion was estimated to be 409.9 and 11.1% at LexieWudan Lake and KekeXili Lake (Zhou et al., 2019). However, more and more studies shows that glaciers in the Karakoram and western Kunlun Mountains are very stable or even exhibited positive mass balance (K ääb et al., 2015<del>Yao et al., 2015</del><del>Yao et al., 2018</del>). For example, K äänab et al (2018) showed that both the two Aru glaciers experienced a slight thickness increase mass gain of 0.2-0.3 m/yr water equivalent (w.e.) since the early 2000s, despite there iwas slight-glacier retreat of 520-460 m. (K ääb et al., 2008). The 2000swas not mainly contributed by the glacier mass changes. Treichler et al (2019) suggested that both the glacier thickening on the western TP and rapid lake growth on the western TP

were mainly attributed to the stepwise increase in precipitation insince the late 1990s. Dramatic increase in precipitation since the 2000s is visible from meteorological station data and reanalysis data (Lei et al., 2014; Treichler et al., 2019). This indicates that rapid lake expansion on the western TP, including Memar Co, mayeantaken-was a response to climate

change, especially climate wetting (Lei and Yang, 2017).

470

**5.2 Potential risk caused by lake expansion on the TP**

DramaieWidespread lake expansion occurred was widely found for most closed lakes on the interior TP during the past two decades (e.g. Lei al., 2014). Lake expansion on the interior TP inundated grassland and infrastructures (e.g. road and bridges)
 in the surrounding area, which not only led to enormous economic loss, but also serious ecological and environmental problems (Yao et al., 2010; Liu et al., 2019; Pei et al., 2019). For example, rapid lake expansion in the northern Tibet inundated a large area of grassland and destroyed infrastructures such as roads and bridges (Yao et al., 2011). A case study occurred in Hol Hil Nature Reserve, where a significant overflow suddenly occurred at Zhuonai Lake (255 km2) in late August 2011 due to continuous expansion since the 2000s. The flood subsequently induced the overflow of Kusai Lake (260 km2) and rapid expansion of the downstream lakes, Haidingnuoer Lake and Salt Lake (Yao et al., 2012; Liu et al., 2019). This sudden process was captured by CryoSat satellite, which shows that there was 12.6 m lake level drop at Zhuonai

- Lake after the outburst (Hwang et al., 2019; Li et al., 2019). The newly formed riverbanks caused by the outburst floodobstructed the traditional migration route of antelopes and had serious ramifications for antelope survival (Pei et al., 2019).The rapid lake expansion of Memar Co ismay also lead to serious ecological problem no exception. The rapid
- 485 expansion of Memar Co may further lead to its combination with Aru Co in near future, which willwill have significant impact on the regional geomorphology and ecosystem. In 2003, the surface elevation of Aru Co (4936.8 m a.s.l) was about 14 m higher than that of Memar Co (4923.2 m a.s.l), as indicated by ICESat satellite altimetry data. In 2014, CryoSat-2 data show that the elevation difference between the two lakes decreased to ~8 m due to continual lake expansion of Memar Co. After the glacier collapses, Memar Co expanded at an accelerated speed and the elevation difference became even smaller. In

- 490
   October 2019, the surface elevation of Memar Co reached 4931.3 m a.s.l and the elevation difference between the two lakes

   decreased to only 5.5 m. According to the increasing rate of 0.5-0.8 m/yr between 2003 and 2019, the surface elevation of

   Memar Co could reach that of Aru Co in 7-11 years. If Memar Co continued to expand as before, the surface elevation of

   Memar Co could reach that of Aru Co in 7-11 years. According to the reconstructed relationship between lake area and lake

   level in section 4.4, when the lake level of Memar Co increases by 5 m, the lake area and water storage will increase by 10.6%
- 495 and 0.65 Gt, relative to 2019.
   As has been shown, Memar Co is a saline lake while Aru Co is a freshwater lake. If the two lakes are merged, lake salinity and ion composition will exchange freely. Memar Co will be diluted while Aru Co will be significantly salted. The habitat of the phytoplankton and zooplankton in the lake will also change significantly in response to changes in lake salinity and ion composition. Therefore, it is necessary to carry out comprehensive monitoring at Aru Co and Memar Co in the next years,
   500 including lake hydrology, metageplagy, unter graphic and conference of a saline salinity and conference of the salinity and conference of the
- 500 including lake hydrology, meteorology, water quality and ecology, etc.

**56 Conclusions**

The fragmented ice mass from of the Aru ice avalanchesglacier collapses on 17 July and 21 September 2016 had almost melted by September 2019. A comprehensive investigation of the two downstream lakes, the outflow lake Aru Co and the

- 505 terminal lake Memar Co, was carried outconducted since 2016, including meteorology, ice mass balance, lake bathymetry, lake level changes, etc. Based on in-situ observation and satellite data, how the two downstream lakes responded to the ice avalanches in the successive years (2016-2019) is evaluated in this study. A comprehensive investigation of the two downstream lakes, the outflow lake Aru Co and the terminal lake Memar Co, was carried out since 2016, including meteorology, ice mass balance, lake bathymetry, lake level changes, etc. How-We found that the ice avalanchesthe two downstream lakes, Aru Co and Memar Co, can significantly affect the two downstream lakesresponded. The impact ofto at
  - least in the following aspects the ice avalanches on thein the successive years (2016-2019) downstream lakes is evaluated in this study based on in situ observation in combination with and satellite data.: The main conclusion is as the following:

Lake bathymetry shows that Aru Co and Memar Co have water storage of 17.9 ×108 m3 and 34.9 ×108 m3, respectively. Although the total volume of the two glacier collapses only accounts for ~8% of the water storage of Aru Co, it exert great

- 515 impacts on the two downstream lakes in terms of lake bathymetry, water temperature and lake level. After Aru-1 glacier collapses, a large amount ofthe -fragmented ice mass\_debris was transportedslid into Aru Co along with a large amount of the debris\_fragmented ice, which generated great surges at Aru Co and further modified the shoreline and bathymetry near the glacier collapse fan. The Aru Co shoreline was pushed inwardsoffshore about 100-120 m along the two sides of the first glacier collapse fan. Lake bathymetry near Aru-1 ice avalanche became much uneven, which is quite different from the adjacent areas. The intruding ice into Aru Co, with an area of ~0.89 km2 and a volume of at least 7.1×106 m3. melted in less
- than two months.

The spread of intrudfloating ice soon spread-over the surface of Aru Co's surface and dramatically lowered lake surface temperature (LST) by 2-4 °C in the first 2 weeks after the firstAru-1 glacier collapse. The Aru Co shoreline was pushed inwards about 100 120 m along the two sides of the first glacier collapse fan. Lake surface temperature at Aru Co decreased significantly by 2 4 °C in the first two weeks after the first glacier collapse. A similar condition also occurred at Memar Co, but its magnitude and duration were much less than that at Aru Co. The dramatic difference of lake surface temperature across Aru Co is investigated before and after Aru-1 glacier collapse. The spatial patterns of lake surface temperature before and after the first glacier collapse shows that lake surface temperature in summer 2016 at the northern Aru Co-is dramatically 
[revised manuscript text omitted]

|---------------------------------------------------------------|-----------------------------------------------|
| Table 1. Annual Fannan, ice availanche metting and lake ieve  | er mercase at memar co between 2010 and 2017  |

| <del>Duration</del>       | <del>Rainfall</del>
( <del>mm)</del> | Ice avalanche melting
(10 6 -m 3 ) | Lake level
increase at
Memar Co (m) | Contribution of ice
melting to lake
expansion (%) |
|---------------------------|-----------------------------------------|-------------------------------------------------------------|-------------------------------------------|---------------------------------------------------------|
| <del>2016.8-2016.10</del> | —                                       | <del>30.6</del>                                             | <del>0.43</del>                           | 4 <del>1.9%</del>                                       |
| <del>2016.10-2017.9</del> | 4 <del>20</del>                         | <del>63.9</del>                                             | <del>1.07</del>                           | <del>34.3%</del>

---

## Author Response (AR2)

**Response to reviewer #1**

General response: We are very grateful to the reviewers' comments, which are very important to further improve the paper. We have revised the paper according to these comments. The replies to these comments are shown as the following:

*The language still needs some fixing that might exceed what the Copernicus editing can solve. Would be really good if some co-authors could check. One of many examples: you call the tsunami by the avalanche impact "surge". This will most likey lead to misunderstanding in a cryopsphere journal where most readers will think of a glacier surge. Language editing will not fix this.*

Response: Thanks for the suggestion. The main text has been polished again by all the co-authors. For example, we use 10 'wave' instead of 'surge' in the main text (e.g. Lines 28, 51, 76, 191) in the revision. 'Floating ice over lake surface' is changed to 'ice floes over the lake surface' throughout the text. If the editor feels the language is not of sufficient quality of a clear understanding, we will request the service of a copy-editing service as none of the co-authors are native speaker.

*you still call 2-4 K lake temp decrease over 1-2 months "dramatic". You need to explain why this is dramatic. What effects*
*does it have?*

Response: We do not use 'dramatic' in the revision. We further address the effect of the abrupt decrease in lake surface temperature in the revision (Lines 335-336). 'Both Terra and Aqua datasets showed that nighttime lake surface temperature at Aru Co decreased abruptly by 2-4 $^{\circ}$C in the first two weeks after the Aru-1 glacier collapse (Fig. 8), which was quite different from normal years and may considerably affect the lake ecosystem.'

*5.2 title: this is not "risk", better "impacts"*

Response: We believe that 'risk' is better than 'impact' because there are too many 'impact' ahead in the main text. Meanwhile, what we want to address is the potential risk of lake expansion.

*The lake temperature decrease of Aru Co should be a mixture of ice floes and real drop in water temperature. How about Memar Co: were there also ice floes on it, or is the temperature drop from lake water only?*

Response: We agree with this. In fact, the lake volume is much larger than the ice intruding into Aru Co. Therefore, the decrease in lake surface temperature is mainly due to ice floes and the meltwater. The decrease in lake surface temperature of Memar Co was also mainly due to ice floes because some of them may flow into Memar Co through the channel between the 30 two lakes (Line 347-348).

*black lines in Fig 3a very hard to read. Try white, use larger font and line widths.*

Response: We have revised Fig 3a and its caption in the revision.

*- Fig 8 and 9, and associated text: without accuracy assessment, and error bars/margins for the LST data, it is hard for reader to interpret the data and tell which differences are statistically significant. In the method section you indicate +- 1 K. Is this value also useful for your site and application? Include this also in the considerations in section 4.6, and add an error bar/margin in Figs 8 and 9.*

Response: We have added error bar for the LST data in Fig 8 and 9. Generally, MODIS derived LST reflects instantaneous
lake surface temperature with an accuracy of $\pm1^{o}C$ (Wan Z. et al., 2013). We also compared MODIS derived LST data with in-situ observation at the shoreline and found that they are generally consistent especially during the post monsoon season (Fig. S5). However, there is some difference in spring and summer between the two dataset. This is because MODIS sensors measured the lake skin temperature at the lake centre while HOBO logger measured lake water temperature at the depth of 30-70 cm at the shoreline (Line 329-330). Until now, there are still few studies about the validation of MODIS derived lake
surface temperature. For example, MODIS derived daytime LST at Qinghai Lake on the northeast TP has been validated to be 1.8 $^{o}$C when using in situ observation at 0.5 m water depth (Xiao et al., 2013).

Xiao, F., Ling, F., Du, Y., Feng, Q., Yan, Y., Chen, H.: Evaluation of spatial-temporal dynamics in surface water temperature of Qinghai Lake from 2001 to 2010 by using MODIS data. J. Arid Land. 5, 452-464, 2013.

**Response to reviewer #2**

General response: We appreciate the reviewers' comments very much. We have revised the paper according to these
comments. Responses to these comments are shown in the following:

Page 3 Line 75 reference (eg. Yao et al., 2010,2018) and next sentence (Yao et a., 2018) Are The two references same?
Response: Thanks for pointing out this error. We have revised the references in the revision (Line 376, 396).

Page 4 Line 105 glaciers(Aru-1 and Aru-2). Two glaciers must be marked in figure 1.
Response: We have marked the two glaciers in Fig. 1.

Page 7 line 225 "the bathymetry survey shows that" could be deleted or changed to "Observation from images show that"
Response: We have revised this in the main text (Line 173).

Page 15 line 475. "For example, rapid lake expansion…for antelope survival (pei et al., 2019). "One sentence is enough. So,
I would suggest that the author abbreviated those sentences.
Response: Thanks for the suggestion. We have simplified this paragraph in the revision (Line 395-397).

[revised manuscript text omitted]